# *EAGLE*: Large-scale Learning of Turbulent Fluid Dynamics with Mesh Transformers

**Steeven Janny**
LIRIS, INSA Lyon, France
steeven.janny@insa-lyon.fr

**Aurélien Béneteau**
SupAero, France
aurelien.beneteau
@student.isae-supaero.fr

**Madiha Nadri**
LAGEPP, Univ. Lyon 1, France
madiha.nadri-wolf@univ-lyon1.fr

**Julie Digne**
LIRIS, CNRS, France
julie.digne@cnrs.fr

**Nicolas Thome**
Sorbonne University, CNRS, ISIR,
Paris, France
nicolas.thome@isir.upmc.fr

**Christian Wolf**
Naver Labs Europe, France
christian.wolf@naverlabs.com

## Abstract

Estimating fluid dynamics is classically done through the simulation and integration of numerical models solving the Navier-Stokes equations, which is computationally complex and time-consuming even on high-end hardware. This is a notoriously hard problem to solve, which has recently been addressed with machine learning, in particular graph neural networks (GNN) and variants trained and evaluated on datasets of static objects in static scenes with fixed geometry. We attempt to go beyond existing work in complexity and introduce a new model, method and benchmark. We propose *EAGLE*, a large-scale dataset of ~1.1 million 2D meshes resulting from simulations of unsteady fluid dynamics caused by a moving flow source interacting with nonlinear scene structure, comprised of 600 different scenes of three different types. To perform future forecasting of pressure and velocity on the challenging *EAGLE* dataset, we introduce a new mesh transformer. It leverages node clustering, graph pooling and global attention to learn long-range dependencies between spatially distant data points without needing a large number of iterations, as existing GNN methods do. We show that our transformer outperforms state-of-the-art performance on, both, existing synthetic and real datasets and on *EAGLE*. Finally, we highlight that our approach learns to attend to airflow, integrating complex information in a single iteration.

## 1 Introduction

Despite consistently being at the center of attention of mathematics and computational physics, solving the Navier-Stokes equations governing fluid mechanics remains an open problem. In the absence of an analytical solution, fluid simulations are obtained by spatially and temporally discretizing differential equations, for instance with the finite volume or finite elements method. These simulations are computationally intensive, take up to several weeks for complex problems and require expert configurations of numerical solvers.

Neural network-based physics simulators may represent a convenient substitute in many ways. Beyond the expected speed gain, their differentiability would allow for direct optimization of fluid mechanics problems (airplane profiles, turbulence resistance, etc.), opening the way to replace traditional trial-and-error approaches. They would also be an alternative for solving complex PDEs where numerical resolution is intractable. Yet, the development of such models is slowed down by the difficulty of collecting data in sufficient quantities to reach generalization. Velocity and pressure field measurements on real world systems require large and expensive devices, and simulation faces the problems described above. For all these reasons, few datasets are freely available for training high-capacity neural networks, and the existing ones either address relatively simple problems which can be simulated in reasonable time and exhibiting very similar behaviors (2D flow on a cylinder, airfoil

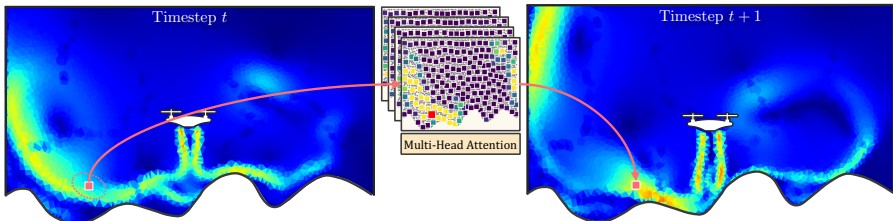

Figure 1: We introduce *EAGLE*, a large-scale dataset for learning complex fluid mechanics, accurately simulating the air flow created by a 2D drone in motion and interacting with scenes of varying 2D geometries. We address the problem through an autoregressive model and self-attention over tokens in a coarser resolution, allowing to integrate long-range dependencies in a single hop — shown in the given example by the attention distributions for ■, which follows the airflow.

(Pfaff et al., 2021; Han et al., 2022)) , or simulations of very high precision, but limited to a few different samples only (Graham et al., 2016; Wu et al., 2017).

In this paper, we introduce *EAGLE*, a large-scale dataset for learning unsteady fluid mechanics. We accurately simulate the airflow produced by a two-dimensional unmanned aerial vehicle (UAV) moving in 2D environments with different boundary geometries. This choice has several benefits. It models the complex ground effect turbulence generated by the airflow of an UAV following a control law, and, up to our knowledge, is thus significantly more challenging than existing datasets. It leads to highly turbulent and non-periodic eddies, and high flow variety, as the different scene geometries generate completely different outcomes. At the same time, the restriction to a 2D scene (similar to existing datasets) makes the problem manageable and allows for large-scale amounts of simulations (∼1.1m meshes). The dataset will be made publically available upon publication.

As a second contribution, we propose a new multi-scale attention-based model, which circumvents the quadratic complexity of multi-head attention by projecting the mesh onto a learned coarser representation yielding fewer but more expressive nodes. Conversely to standard approaches based on graph neural networks, we show that our model dynamically adapts to the airflow in the scene by focusing attention not only locally, but also over larger distances. More importantly, attention for specific heads seems to align with the predicted airflow, providing evidence of the capacity of the model to integrate long range dependencies in a single hop — see Figure 1. We evaluate the method on several datasets and achieve state-of-the-art performance on two public fluid mechanics datasets (Cylinder-Flow, (Pfaff et al., 2021) and Scalar-Flow (Eckert et al., 2019)), and on *EAGLE*.

## 2 RELATED WORK

**Fluids datasets for deep learning** – are challenging to produce in many ways. Real world measurement is complicated, requiring complex velocimetry devices (Wang et al., 2020; Discetti & Coletti, 2018; Erichson et al., 2020). Remarkably, (Eckert et al., 2019; De Bézenac et al., 2019) leverage alignment with numerical simulation to extrapolate precise GT flows on real world phenomena (smoke clouds and sea surface temperature). Fortunately, accurate simulation data can by acquired through several solvers, ranging from computer graphics-oriented simulators (Takahashi et al., 2021; Pfaff & Thuerey, 2016) to accurate computational fluid dynamics solver (OpenFOAM©, Ansys© Fluent, ...). A large body of work (Chen et al., 2021a; Pfaff et al., 2021; Han et al., 2022; Stachenfeld et al., 2021; Pfaff et al., 2021) introduces synthetic datasets limited to simple tasks, such as 2D flow past a cylinder. *EAGLE* falls into this synthetic category, but differs in two main points: (a) simulations rely on hundreds of procedurally generated scene configurations, requiring several weeks of calculations on a high-performance computer, and (b) we used an engineer-grade fluid solver with demanding turbulence model and a fine domain discretization. For a comparison, see table 1.

**Learning of fluid dynamics** – is mainly addressed with message passing networks. Recent work focuses in particular on smoothed-particle hydrodynamics (SPH) (Shao et al., 2022; Ummenhofer et al., 2020; Shlomi et al., 2021; Li et al., 2019; Allen et al., 2022), somehow related to a Lagrangian representation of fluids. Sanchez-Gonzalez et al. (2020) proposes to chain graph neural networks in an Encode-Process-Decode pipeline to learn interactions between particles. SPH simulations are

| Dataset | | Size | Public | Dyn. Scene | Dyn. Mesh | # nodes (avg) | # of meas. |
|---|---|---|---|---|---|---|---|
| Pfaff et al. (2021) | CylinderFlow | 15Gb | ✓ | ✗ | ✗ | 1,885 | 0.72M |
| | AirFoil | 56Gb | | | | 5,233 | 0.72M |
| | KS Equation | | | | | 64 | 1,200 |
| Stachenfeld et al. (2021) | Incomp. Dec. | N.A | ✗ | ✗ | ✗(Grid) | 2,304 | 210 |
| | Comp. Dec. | | | | | 32,768 | 35 |
| | Rad. Cooling | | | | | 32,768 | 30 |
| Han et al. (2022) | Vascular Flow | N.A. | ✗ | ✗ | ✓ | 7,561 | 5,250 |
| Eckert et al. (2019) | | 351Gb | ✓ | ✗ | ✗(Grid) | 1.7M | 0.015M |
| De Bézenac et al. (2019) | SST | N.A | ✓ | ✗ | ✗(Grid) | 4,096 | 0.1M |
| *EAGLE*(Ours) | | **270Gb** | **✓** | **✓** | **✓** | **3,388** | **1.18M** |

Table 1: Fluid mechanics datasets in the literature. To the best of our knowledge, *EAGLE* is the first dataset of such scale, complexity and variety. Smaller-scale datasets such as Li et al. (2008); Wu et al. (2017) have been excluded, as they favor simulation accuracy over size. The datasets in Stachenfeld et al. (2021) are not public, but can be reproduced from the information in the paper.

very suitable for applications with reasonable number of particles but larger-scale simulations (vehicle aerodynamic profile, sea flows, etc.) remain out of scope. In fluid mechanics, Eulerian representations are more regularly used, where the flow of quantities are studied on fixed spatial cells. The proximity to images makes uniform grids appealing, which lead to the usage of convolutional networks for simulation and learning (De Bézenac et al., 2019; Ravuri et al., 2021; Ren et al., 2022; Liu et al., 2022; Le Guen & Thome, 2020). For instance, Stachenfeld et al. (2021) takes the principles introduced in Sanchez-Gonzalez et al. (2020) applied to uniform grids for the prediction of turbulent phenomena. However, uniform grids suffer from limitations that hinder their generalized use: they adapt poorly to complex geometries, especially strongly curved spatial domains and their spatial resolution is fixed, requiring a large number of cells for a given precision.

**Deep Learning on non-uniform meshes** – are a convenient way of solving the issues raised by uniform grids. Nodes can then be sparser in some areas and denser in areas of interest. Graph networks (Battaglia et al., 2016) are well suited for this type of structure. The task was notably introduced in Pfaff et al. (2021) with MeshGraphNet, an Encode-Process-Decode pipeline solving mesh-based physics problems. (Lienen & Günnemann, 2022) introduced a graph network structure algorithmically aligned with the finite element method and show good performances on several public datasets. Close to our work, Han et al. (2022) leverages temporal attention mechanism on a coarser mesh to enhance forecasting accuracy over longer horizon. In contrast, our model is based on a spatial transformer, allowing a node to communicate not only with its neighbors but also over greater distances by dynamically adapting attention to the airflow.

*EAGLE* is comprised of fine-grained fluid simulations defined on irregular triangle meshes, which we argue is more suited to a broader range of applications than regular grids and thus more representative of industrial standards. Compared to grid-based datasets De Bézenac et al. (2019); Stachenfeld et al. (2021), irregular meshes provide better control over the spatial resolution, allowing for finer discretization near sensitive areas. This property is clearly established for most fluid mechanics solvers (Versteeg & Malalasekera, 2007) and seems to transfer well to simulators based on machine learning (Pfaff et al. (2021)). However, using triangular meshes with neural networks is not as straightforward as regular grids. Geometric deep learning (Bronstein et al., 2021) and graph networks (Battaglia et al., 2018) have established known baselines but this remains an active domain of research. Existing datasets focus on well-studied tasks such as the flow past an object (Chen et al., 2021a; Pfaff et al., 2021) or turbulent flow on an airfoil (Thuerey et al., 2020; Sekar et al., 2019). These are well studied problems, for some of which analytical solutions exist, and they rely on a large body of work from the physics community. However, the generated flows, while being turbulent, are merely steady or periodic despite variations in the geometry. With *EAGLE*, we propose a complex task, with convoluted, unsteady and turbulent air flow with minimal resemblance across each simulation.

## 3 THE *EAGLE* DATASET AND BENCHMARK

**Purpose** – we built *EAGLE* in order to meet a growing need for a fluid mechanics dataset in accordance with the methods used in engineering, i.e. reasoning on irregular meshes. To significantly increase the complexity of the simulations compared to existing datasets, we propose a proxy task consisting in studying the airflow produced by a dynamically moving UAV in many scenes with

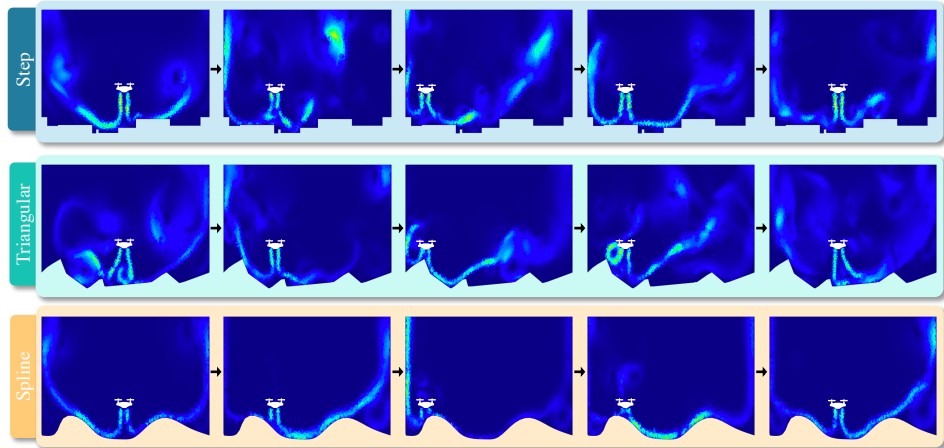

Figure 2: Velocity field norm over time for three episodes, one for each geometry type. Turbulence is significantly different from one simulation to another and strongly depends on the ground surface.

variable geometry. This is motivated by the highly non-steady turbulent outcomes that this task generates, yielding challenging airflow to be forecasted. Particular attention has also been paid to the practical usability of *EAGLE* with respect to the state-of-the-art in future forecasting of fluid dynamics by controlling the number of mesh points, and limiting confounders variables to a moderate amount (i.e. scene geometry and drone trajectory).

**Simulation and task definition** – we simulate the complex airflow generated by a 2D unmanned aerial vehicle maneuvering in 2D scenes with varying floor profile. While the scene geometry varies, the UAV trajectory is constant: the UAV starts in the center of the scene and navigates, hovering near the floor surface. During the flight, the two propellers generate high-paced air flows interacting with each other and with the structure of the scene, causing convoluted turbulence. To produce a wide variety of different outcomes, we procedurally generate a large number of floor profiles by interpolating a set of randomly sampled points within a certain range. The choice of interpolation order induces drastically different floor profiles, and therefore distinct outcomes from one simulation to another. *EAGLE* contains three main types of geometry depending on the type of interpolation (see Figure 2): **(i) Step**: surface points are connected using step functions (zero-order interpolation), which produces very stiff angles with drastic changes of the air flow when the UAV hovers over a step. **(ii) Triangular**: surface points are connected using linear functions (first-order interpolation), causing the appearance of many small vortices at different location in the scene. **(iii) Spline**: surface points are connected using spline functions with smooth boundary, causing long and fast trails of air, occasionally generating complex vortices.

*EAGLE* contains about 600 different geometries (200 geometries of each type) corresponding to roughly 1,200 flight simulations (one geometry gives two flight simulations depending on whether the drone is going to the right or to the left of the scene), performed at 30 fps over 33 seconds, resulting in 990 time steps per simulation. Physically plausible UAV trajectories are obtained through MPC control of a (flow agnostic) dynamical system we design for a 2D drone. More details and statistics are available in appendix A.

We simulated the temporal evolution of the velocity field as well as the pressure field (both static and dynamic) defined over the entire domain. Due to source motion, the triangle mesh on which these fields are defined need to be dynamically adapted to the evolving scene geometry. More formally, the mesh is a valued dynamical graph $\mathcal{M}^t = \left(\mathcal{N}^t, \mathcal{E}^t, \mathcal{V}^t, \mathcal{P}^t\right)$ where $\mathcal{N}$ is the set of nodes, $\mathcal{E}$ the edges, $\mathcal{V}$ is a field of velocity vectors and $\mathcal{P}$ is a field of scalar pressure values. Both physical quantities are expressed at node level. Note that the dynamical mesh is completely flow-agnostic, thus no information about the flow can be extrapolated directly from the future node positions. Time dependency will be omitted when possible for sake of readability.

**Numerical simulations** – were carried out using the software Ansys© Fluent, which solves the Reynolds Averaged Navier-Stokes equations of the Reynolds stress model. It uses five equations to model turbulence, more accurate than standard $k$-$\epsilon$ or $k$-$\omega$ models (two equations). This resulted in

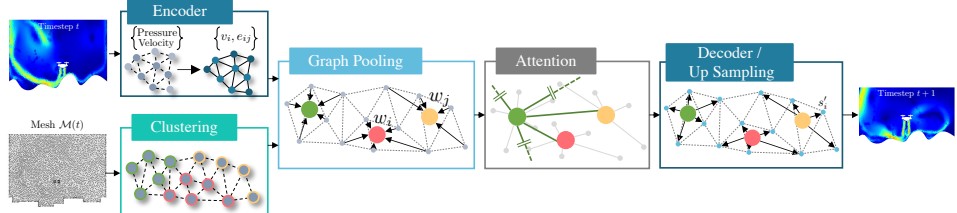

Figure 3: The mesh transformer encodes the input mesh node values (positions, pressure and velocity), reduces the spatial resolution through clustering + graph pooling, and performs multi-head self-attention on the coarser level of cluster centers. A decoder upsamples the token embeddings to the original resolution and predicts pressure and velocity at time step $t + 1$.

3.9TB of raw data with $\sim$162,760 control points per mesh. We down-sampled this to 3,388 points in average, and compressed it to 270GB. Details and illustrations are given in appendix A.

**Task** – for what follows, we define $x_i$ as the 2D position of node $i$, $v_i$ its velocity, $p_i$ pressure and $n_i$ is the node type, which indicates if the node belongs to a wall, an input or an output boundary. We are interested in the following task: given the complete simulation state at time $t$, namely $\mathcal{M}^t$, as well as future mesh geometry $\mathcal{N}^{t+h}, \mathcal{E}^{t+h}$, forecast the future velocity and pressure fields $\hat{\mathcal{V}}^{t+h}, \hat{\mathcal{P}}^{t+h}$, i.e. for all positions $i$ we predict $\hat{v}_i^{t+h}, \hat{p}_i^{t+h}$ over a horizon $h$. Importantly, we consider the dynamical re-meshing step $\mathcal{N}^t \rightarrow \mathcal{N}^{t+h}$ to be known during inference and thus is not required to be forecasted.

## 4 LEARNING UNSTEADY AIRFLOW

Accurate flow estimations require data on a certain minimum spatial and temporal scale. Deviations from optimal resolutions, i.e. data sampled with lower spatial resolutions or lower frame rates, are typically very hard to compensate through models of higher complexity, in particular when the estimation is carried out through numerical simulations with an analytical model. The premise of our work is that machine learning can compensate loss in resolution by picking up longer rate regularities in the data, trading data resolution for complexity in the modeled interactions. Predicting the outcome for a given mesh position may therefore require information from a larger neighborhood, whose size can depend on factors like resolution, compressibility, Reynolds number etc.

Regularities and interactions on meshes and graphs have classically been modeled with probabilistic graphical models (MRFs (Geman & Geman, 1984), CRFs (Lafferty et al., 2001), RBMs (Smolensky, 1986) etc.), and in the DL era through geometric DL (Bronstein et al., 2021) and graph networks (Battaglia et al., 2018), or through deep energy-based models. These models can capture long-range dependencies between distant nodes, but need to exploit them through multiple iterations. In this work we argue for the benefits of transformers and self-attention Vaswani et al. (2017), which in principle are capable of integrating long-range interactions in a single step.

However, the quadratic complexity of transformers in terms of number of tokens makes its direct application to large meshes expensive. While low-complexity variants do exist, e.g. (Katharopoulos et al., 2020), we propose a different *Ansatz*, shown in Figure 3: we propose to combine graph clustering and learned graph pooling to perform full attention on a coarser scale with higher-dimensional node embedding. This allows the dot-product similarity of the transformer model — which is at the heart of the crucial attention operations — to operate on a semantic representation instead of on raw input signals, similar to the settings in other applications. In NLP, attention typically operates on word embeddings Vaswani et al. (2017), and in vision either on patch embeddings Dosovitskiy et al. (2021) or on convolutional feature map cells Wang et al. (2018). In the sequel, we present the main modules of our model; further details are given in appendix B

**Offline Clustering** – we down-scale mesh resolution through geometric clustering, which is independent of the forecasting operations and therefore pre-computed offline. A modified k-means clustering is applied to the vertices $\mathcal{N}^t$ of each time step and creates clusters with a constant number of nodes, details are given in appendix B.1. The advantages are twofold: (a) the irregularity and adaptive resolution of the original mesh is preserved, as high density region will require more clusters, and (b)

constant cluster sizes facilitate parallelization and allow to speed up computations. In what follows, let $C_k$ be the $k^{th}$ cluster computed on mesh $\mathcal{M}^t$.

**Encoder** – the initial mesh $\mathcal{M}^t$ is converted into a graph $\mathcal{G}$ using the encoder in Pfaff et al. (2021). More precisely, node and edge features are computed using MLPs $\phi_{\text{node}}$ and $\phi_{\text{edge}}$, giving

$$\eta_i^1 = \phi_{\text{node}}(v_i, p_i, n_i), \quad e_{ij}^1 = \phi_{\text{edge}}(x_i - x_j, \|x_i - x_j\|). \tag{1}$$

The encoder also computes an appropriate positional encoding based upon spectral projection $F(x)$. We also leverage the local position of each node in its cluster. Let $\bar{x}_k$ be the barycenter of cluster $C_k$, then the local encoding of node $i$ belonging to cluster $k$ is the concatenation $f_i = [F(x_i)\ F(\bar{x}_k - x_i)]^T$. Finally, a series of $L$ Graph Neural Networks (GNN) extracts local features through message passing:

$$e_{ij}^{\ell+1} = e_{ij}^{\ell} + \underbrace{\psi_{\text{edge}}^{\ell}\left([\eta_i^{\ell}\ f_i], [\eta_j^{\ell}\ f_j], e_{ij}^{\ell}\right)}_{\varepsilon_{ij}}, \qquad \eta_i^{\ell+1} = \eta_i^{\ell} + \psi_{\text{node}}^{\ell}\left([\eta_i^{\ell}\ f_i], \sum_j \varepsilon_{ij}\right). \tag{2}$$

The superscript $\ell$ indicates the layer, and $\psi_{\text{node}}^{\ell}$ and $\psi_{\text{edge}}^{\ell}$ are MLPs which encode nodes and edges, respectively. The exact architecture hyper-parameters are given in appendix B. For the sake of readability, in whats follows, we will note $\eta_i = \eta_i^L$ and $e_{ij} = e_{ij}^L$.

**Graph Pooling** – summarizes the state of the nodes of the same cluster $C_k$ in a single high-dimensional embedding $w_k$ on which the main neural processor will reason. This is performed with a Gated Recurrent Unit (GRU) Cho et al. (2014) where the individual nodes are integrated sequentially in a random order. This allows to learn a more complex integration of features than a sum. Given an inital GRU state $h^0 = 0$, node embeddings are integrated iteratively, indicated by superscript $n$,

$$h_k^{n+1} = \text{GRU}([\eta_i, f_i], h_k^n), \ i \in C_k, \ w_k = \phi_{\text{cluster}}(h_k^N), \tag{3}$$

where $N = |C_k|$ and $\phi_{\text{cluster}}$ is an MLP. GRU$(\cdot)$ denotes the update equations of a GRU, where we omitted gating functions from the notation. The resulting set of cluster embeddings $\mathcal{W} = w_k|_{k=1..K}$ significantly reduces the spatial complexity of the mesh.

**Attention Module** – consists of a transformer with $M$ layers of multi-head attention (MHA) Vaswani et al. (2017) working on the embeddings $\mathcal{W}$ of the coarse graph. Setting $w_k^1 = w_k$, we get for layer $m$:
$$w_k^{m+1} = \text{MHA}\left(Q = [w_k^m\ F(\bar{x}_k)], K = \mathcal{W}, = \mathcal{W}\right), \tag{4}$$

where Q, K and V are, respectively, the query, key and value mappings of a transformer. We refer to Vaswani et al. (2017) for the details of multi-head attention, denoted as MHA$(\cdot)$.

**Decoder** – the output of the attention module is calculated on the coarse scale, one embedding per cluster. The decoder upsamples the representation and outputs the future pressure and velocity field on the original mesh. This upsampling is done by taking the original node embedding $\eta_i$ and concatenating with the cluster embedding $w_k^M$, followed by the application of a GNN, whose role is to take the information produced on a coarser level and correctly distribute it over the nodes $i$. To this end, the GNN has access to the positional encoding of the node, which is also concatenated:

$$\begin{cases} \hat{v}^{t+1} = v^t + \delta_v \\ \hat{p}^{t+1} = p^t + \delta_p \end{cases}, \ (\delta_v, \delta_p) = \text{GNN}\left([\eta_i\ w_k^M\ f_i]\right), \tag{5}$$

where $i \in C_k$ and GNN$(\cdot)$ is the graph network variant described in equation (2), parameters are not shared. Our model is trained end-to-end, minimizing the forecasting error over horizon $H$ where $\alpha$ balances the importance of pressure field over velocity field:

$$\mathcal{L} = \sum_{i=1}^{H} \text{MSE}\left(v(t+i), \hat{v}(t+i)\right) + \alpha \sum_{i=1}^{H} \text{MSE}\left(p(t+i), \hat{p}(t+i)\right). \tag{6}$$

## 5 EXPERIMENTS

We compare our method against three competing methods for physical reasoning: **MeshGraphNet** (Pfaff et al., 2021) (MGN) is a Graph Neural Network based model that relies on multiple chained message passing layers. **GAT** is based upon MGN where the GNNs interactions are replaced by

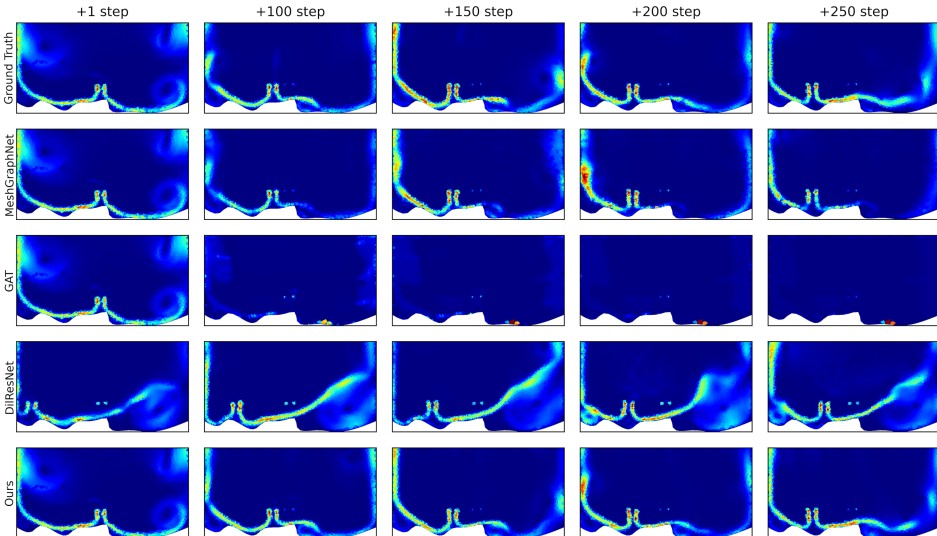

Figure 4: Qualitative comparisons with the state of the art on *EAGLE*. We color code the norm of the velocity field.

| Dataset | Cylinder Flow | | | Scalar Flow | | | *EAGLE* | | |
|---|---|---|---|---|---|---|---|---|---|
| Horizon | +1 | +50 | +250 | +1 | +50 | +100 | +1 | +50 | +250 |
| MeshGraphNet | 0.0058 | 0.0405 | 0.0792 | 0.0374 | 0.3193 | 0.5944 | 0.0916 | 0.5698 | 0.9896 |
| GAT | 0.0112 | 0.1774 | 1.3336 | 0.0434 | 0.3991 | 0.6414 | 7.5587 | 19.260 | 26.867 |
| DilResNet | 0.0429 | 0.0626 | 0.1295 | 0.0372 | 0.2212 | 0.3975 | 0.2987 | 0.5650 | 0.8944 |
| **Ours** | **0.0030** | **0.0221** | **0.0735** | **0.0128** | **0.0869** | **0.1467** | **0.0811** | **0.3495** | **0.6357** |

Table 2: Norm. RMSE (velocity and pressure) for our model (cluster size = 10) and the baselines.

graph attention transformers (Veličković et al., 2017). Compared to our mesh transformer, here attention is computed over the one-ring of each node only. **DilResNet** (DRN) Stachenfeld et al. (2021) differs from the other models as it does not reason over non uniform meshes, but instead uses dilated convolution layers to perform predictions on regular grids. To evaluate this model on *EAGLE*, we interpolate grid-based simulation over the original mesh, see appendix A.2. During validation and testing, we project airflow back from the grid to the original mesh in order to compute comparable metrics. All baselines have been adapted to the dataset using hyperparameter sweeps, which mostly lead to increases in capacity, explained by *EAGLE*'s complexity.

We also compare to two other datasets: **Cylinder-Flow** (Pfaff et al., 2021) simulates the airflow behind a cylinder with different radius and positions. This setup produces turbulent yet periodic airflow corresponding to Karman vortex. **Scalar-Flow** (Eckert et al., 2019) contains real world measurements of smoke cloud. This dataset is built using velocimetry measurements combined with numerical simulation aligned with the observations. Following Lienen & Günnemann (2022); Kohl et al. (2020), we reduce the data to 2D grid-based simulation by averaging along the *x*-direction.

We evaluate all models reporting the sum of the root mean squared error (N-RMSE) on both pressure and velocity fields, which have been normalized wrt to the training set (centered and reduced), and we provide finer-grained metrics in appendix C.1.

**Existing datasets** – show little success to discriminate the performances of fluid mechanics models (see table 2). On Cylinder-Flow, both ours and MeshGraphNet reach near perfect forecasting accuracy. Qualitatively, flow fields are hardly distinguishable from the ground truth at least for the considered horizon (see appendix C.2). As stated in the previous sections, this dataset is a great task to validate fluid simulators, but may be considered as saturated. Scalar-Flow is a much more challenging benchmark, as these real world measurements are limited in resolution and quantity. Our model obtains good quantitative results, especially on a longer horizon, showing robustness to error accumulation during auto-regressive forecasting. Yet, no model achieved visually satisfactory results, the predictions remain blurry and leave room for improvements (cf figure in appendix).

**Comparisons with the state-of-the-art** – are more clearly assessed on *EAGLE*. Our model gives excellent results and outperforms competing baselines. It succeeds in forecasting turbulent eddies even after a long prediction horizon. Our model outperforms MeshGraphNet, which provides evi-

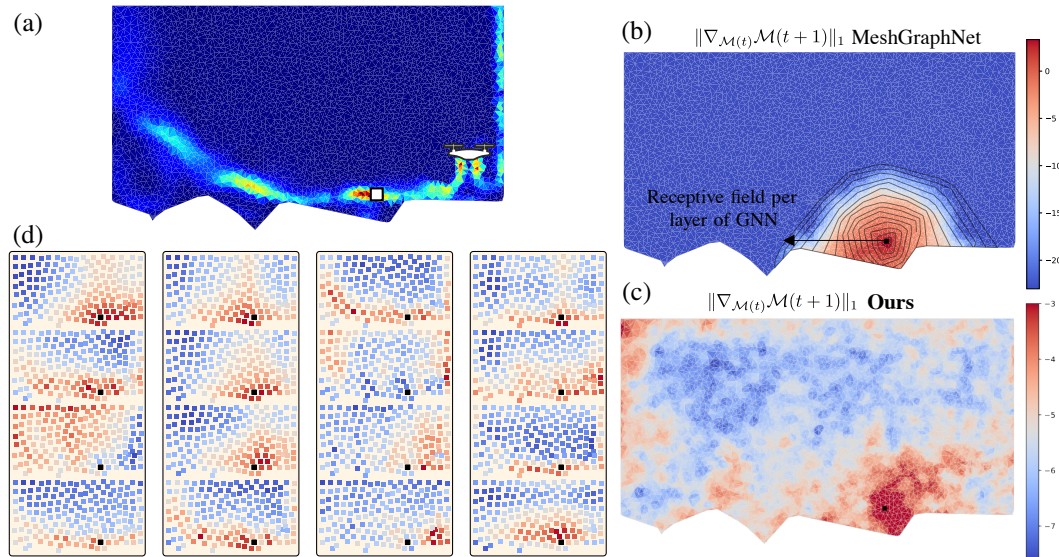

Figure 5: **Locality of reasoning**. (a) velocity of an example flow and a selected point □; (b): The receptive field for this point for the MeshGraphNet model (Pfaff et al., 2021) is restricted to a local neighborhood, also illustrated through the overlaid gradients $||\nabla_{\mathcal{M}(t)}\mathcal{M}(t+1)||_1$. (c): the receptive field of our method covers the whole field and the gradients indicate that this liberty is exploited; (d) the attention distributions for point □, certain maps correlate with airflow. Attention maps can be explored interactively using the online tool at https://eagle-dataset.github.io.

dence for the interest in modeling long-range interactions with self-attention. GAT seems to struggle on our challenging dataset. The required increase in capacity was difficult to do for this resource hungry model, we failed even on the 40GB A100 GPUs of a high-end Nvidia DGX.

DilResNet shows competitive performances on *EAGLE*, consistent with the claims of the original paper. However, it fails to predict details of the vortices (cf. Figure 4). This model leverages grid-based data, hence was trained on a voxelled simulation, finally projected back on the triangular mesh during testing. This requires precaution in assessment. We try to limit projection error by setting images to contains ten times more pixels than nodes in the actual mesh. Yet, even at that scale, we measure that the reconstruction error represents roughly a third of the final N-RMSE. This points out that grid-based are not suited for complex fluid problem such as *EAGLE*, which require finer spatial resolution near sensitive areas. We expose failure cases in appendix C.3.

**Self-attention** – is a key feature in our models, as shown in Figure 5b and c, which plots the gradient intensity of a selected predicted point □ situated on the trail wrt. to all input points, for fixed trained model weights. MeshGraphNet is inherently limited to a neighborhood determined by the number of chained GNNs, the receptive field, which is represented as concentric black circles overlaid over the gradients. In contrast, our model is not spatially limited and can exchange information across the entire scene, even possible in a single step. The gradients show that this liberty is exploited.

| Ablation | Clustering | N-RMSE (+250) |
|---|---|---|
| GNN | 20 | 1.3484 |
| One-ring | 1 | 1.0258 |
| | 20 | 0.7976 |
| Average | 1 | 0.7876 |
| | 20 | 0.7797 |
| Ours | 20 | **0.6572** |

Figure 6: **Ablations:** *GNN* replaces global attention by a set of *L* GNNs on the coarser mesh. *One-ring* constrains attention to the one-ring. *Average* forces uniform attention.

In the same figure we also show the attention maps, per head and layer, for the selected point □ near the main trail in Figure 5d. Interestingly, our model discovers to attend not only to the neighborhood (as a GNN would), but also to much farther areas. More importantly, we observe that certain heads explicitly (and dynamically) focus on the airflow, which provides evidence that attention is guided by the regularities in the input data. We released an online tool allowing interactive visualization and exploration of attention and predictions, available at https://eagle-dataset.github.io.

**Ablation studies** – indicate how global attention impacts performance: (a) closer to MeshGraphNet, we replace the attention layers by GNNs operating on the coarser mesh, allowing message passing between nearest cluster only; (b) we limit the receptive field of MHA to the **one-ring** of the cluster;

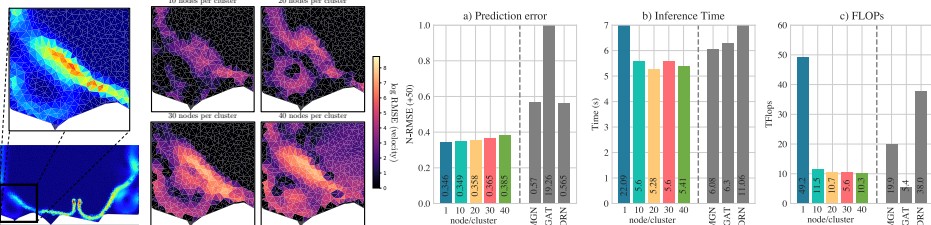

Figure 7: **Impact of cluster size**. Left: We color code RMSE in logarithmic scale on the velocity field near a relatively turbulent area at a horizon of $h$=400 steps of forecasting. Right: Error (+50), inference time and FLOPs for different cluster sizes and the baselines.

| Model | Ours | | | | MGN | | | |
|---|---|---|---|---|---|---|---|---|
| **Ablated Geometry** | Stp | Spl | Tri | ∅ | Stp | Spl | Tri | ∅ |
| N-RMSE (+250)    Stp | 0.927 | 0.865 | 1.132 | **0.828** | 2.062 | 1.236 | 1.347 | 1.116 |
| Spl | 0.595 | 0.584 | 0.857 | **0.488** | 1.257 | 0.941 | 1.037 | 0.807 |
| Tri | 0.730 | 0.732 | 1.049 | **0.647** | 1.685 | 1.100 | 1.131 | 1.037 |

Table 3: **Generalization to unseen geometries:** we evaluate our model and MeshGraphNet in different setups, evaluating on all geometry types but removing one from training. Our model shows satisfactory generalization, also highlighting the complementarity of each simulation type.

(c) we enforce uniform attention by replacing it with an **average** operation. As shown in table 6, attention is a key design choice. Disabling attention to distant points has a negative impact on RMSE, indicating that the model leverages efficient long-range dependencies. Agnostic attention to the entire scene is not pertinent either: to be effective, attention needs to dynamically adapt to the predicted airflow. We also conduct a study on generalization to down-sampled meshes in appendix C.4.

**The role of clustering** – is to summarize a set of nodes into a unique feature vector. Arguably, with bigger clusters, more node-wise information must be aggregated in a finite dimensional vector. We indeed observe a slight increase in N-RMSE when the cluster size increases (Figure 7a). Nonetheless, our model appears to be robust to even aggressive graph clustering as the drop remains limited and still outperforms the baselines. A qualitative illustration is shown figure 7 (left), where we simulate the flow up to 400 time-steps forward and observe the error on a relatively turbulent region. Clustering also acts on the complexity of our model by reducing the number of tokens on which attention is computed. We measure a significant decrease in inference time and number of operations (FLOPs) even when we limit clusters to a small size (Figure 7b and c).

**Generalization experiments** – highlight the complementarity of the geometry types in *EAGLE*, since the removal of one geometry in the training set impacts the performances on the others. Mesh-GraphNet suffers the most, resulting in a drop ranging from 10% in average (ablation of Spline) to 67% (ablation of Step). On our model, the performance losses are limited for the ablation of Step and Spline. The most challenging geometry is arguably Triangular, as the ground profile tends to generate more turbulent and convoluted flows.

# 6 CONCLUSION[1]

We presented a new large-scale dataset for deep learning in fluid mechanics. *EAGLE* contains accurate simulations of the turbulent airflow generated by a flying drone in different scenes. Simulations are unsteady, highly turbulent and defined on dynamic meshes, which represents a real challenge for existing models. To the best of our knowledge, we released the first publicly available dataset of this scale, complexity, precision and variety. We proposed a new model leveraging mesh transformers to efficiently capture long distance dependencies on a coarser scale. Through graph pooling, we show that our model reduces the complexity of multi-head attention and outperforms the competing state-of-the-art on, both, existing datasets and *EAGLE*. We showed across various ablations and illustration that global attention is a key design choice and observed that the model naturally attends to airflow. Future work will investigate the impact of implicit representations on fluid mechanics, and we discuss the possibility of an extension to 3D data in appendix D.

[1]**Acknowledgements** – we recognize support through French grants "*Delicio*" (ANR-19-CE23-0006) of call CE23 "*Intelligence Artificielle*" and "*Remember*" (ANR-20-CHIA0018), of call "*Chaires IA hors centres*".

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

# Appendix

## A  DATASET DETAILS

### A.1  STRUCTURE AND POST-PROCESSING

The *EAGLE* dataset is composed of exactly 1,184 simulations of 990 time-steps (33 seconds at 30 fps). Scene geometries are arranged in three categories based on the order of the interpolation used to generate the ground structure: 197 **Step** scenes, 199 **Triangular** and 196 **Spline**. A geometry gives two simulations depending on whether the drone is crossing the left or the right part of the scene. A proper train/valid/test splitting is provided ensuring that each geometry type is equally represented. The train split contains 948 simulations, while test and valid splits each contain 118 simulations.

**Simulation details** – The scene is described as a 5 m×2.5 m 2D surface. Wall boundary conditions (zero velocity) are applied to the frontiers, except for the top edge, which is an outlet (zero diffusion of flow variables). The propellers is modeled as two squares starting in the middle of the scene, with wall boundary conditions on the left, right and top edges, and inlet condition for the bottom edge (normal velocity of intensity proportional to the rotation speed of the propeller). We mesh the scene with triangular cells of an average size of 15 mm, and add inflation near wall boundaries. We let the simulator updates the mesh during time with default parameters.

**Drone trajectory control** – has received special care, and is obtained using model predictive control (MPC) of a dynamical model of a 2D drone allowing realistic trajectory tracking. The model is obtained by constraining the dynamics of a 3D drone model (Romero et al., 2022) to motion in a 2D plane and reducing the number of rotors to two. The drone can therefore move along the axis $x$ and $y$, and pivot around the $z$-axis perpendicular to the simulation plane as follows:

$$\begin{cases} \ddot{x} & = -K_1(\Omega_1^2 + \Omega_2^2)\sin(\theta) + K_2(\Omega_1 + \Omega_2)\dot{x} \\ \ddot{y} & = K_1(\Omega_1^2 + \Omega_2^2)\cos(\theta) - g + K_2(\Omega_1 + \Omega_2)\dot{y} \\ \ddot{\theta} & = K_3(\Omega_2^2 - \Omega_1^2), \end{cases} \tag{7}$$

where $x, y$ is the 2D position of the drone and $\theta$ its orientation, $\Omega_1$ and $\Omega_2$ the left/right propeller rotation speed, $g = 9.81\text{m/s}$ is acceleration (gravity), and $K_1 = 10^{-4}$, $K_2 = 5\times10^{-5}$, $K_3 = 5.5\times10^{-3}$ are physical constants depending on drone geometry. The resulting trajectories represent physically plausible outcomes, taking into account inertia and gravity.

**Mesh down-sampling** – consists in simplifying the raw simulation data, as they are not suitable for direct deep learning applications, and require post-processing (see Figure 8a). The simulation software leverages a very fine-grained mesh dynamically updated in order to accurately solve the Navier-Stokes equations. The main step thus consists in simplifying the mesh to a reasonable number of nodes. Formally, our goal is to construct a new coarser mesh $(\mathcal{N}(t)', \mathcal{E}(t)')$ based upon the raw mesh proposed by the simulation software $(\mathcal{N}(t), \mathcal{E}(t))$. To cope with the dynamic nature of the simulation mesh, our approach consists in dividing the target node set into a static and a dynamic part $\mathcal{N}(t)' = \mathcal{S} + \mathcal{D}(t)$.

- **The static mesh** is obtained by subsampling the simulation point cloud using Poisson Disk Sampling ((Cook, 1986)). However, the spatial density of $\mathcal{N}(t)$ evolves over time (certain areas of space are more densely populated at the end of the simulation than at the start). To preserve finer resolution near relevant regions, we thus concatenated 5 regularly spaced point clouds $\mathcal{N}(t_k)$ into a single set. We then sub-sample the resulting set by randomly selecting a point, and deleting all neighbors in a sphere of radius $R$ around the chosen point. This operation is repeated until no more point is at a distance less than $R$ from another. We used an adaptive radius $R$ correlated to the density map: when the original point cloud is dense, the radius is smaller. Conversely, the radius increases in sparse areas. An example of the density map is provided in Figure 8b.

- **The dynamic mesh** is mandatory to track drone motion accurately. We therefore complete the static mesh with a dynamical part that follows the boundaries of the UAV. To do so, we used

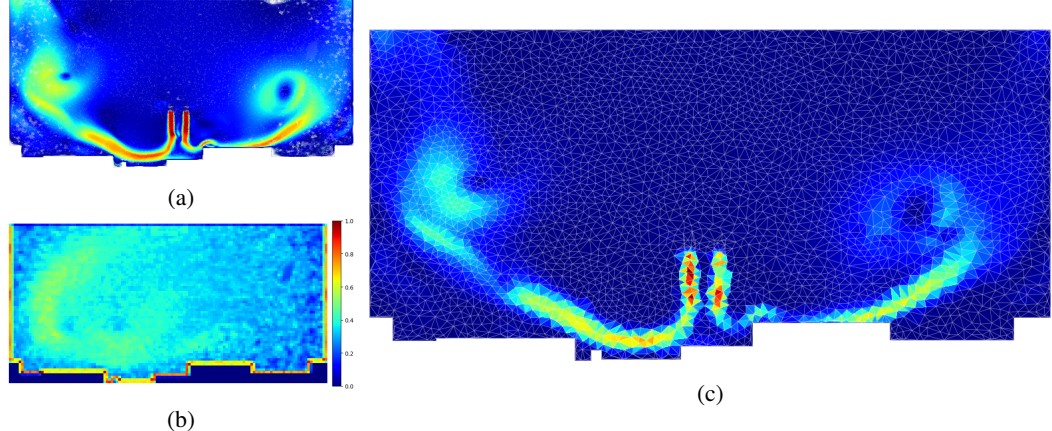

Figure 8: (a) sample of raw simulation measurements obtained on a high resolution mesh. This single snapshot contains 158,961 nodes. (b) example of node density map controlling the sampling disk radius. The raw mesh is more dense near the boundaries and on the left side, as this sample is taken from a simulation where the drone explores the left region of the scene. (c) Mesh simulation at final resolution. We drastically simplified the mesh while maintaining a satisfactory level of details.

the ground truth trajectory to track drone position and orientation across time and extrapolate bounding boxes, which are then transformed into point clouds by sub-dividing the box into several points.

Finally, the edge set $\mathcal{E}'(t)$ is computed using constrained Delaunay triangulation to prevent triangles to spawn outside of the domain. Once $(\mathcal{N}(t)', \mathcal{E}(t)')$ has been computed, we evaluate the pressure and velocity field $\mathcal{V}(t), \mathcal{P}(t)$ on the nodes by averaging the three nearest points in raw simulation data. We illustrate the final result in figure 8c. Better mesh simplification algorithm exists, notably minimizing the interpolation error, yet such algorithms rely on the simulated flow to compute the mesh, which may embed unwanted biases or shortcuts in the mesh geometry.

## A.2 GRID BASED DATASET

One of the baselines, DilResNet (Stachenfeld et al., 2021), relies on convolutional layers for future forecasting of turbulent flows, and therefore requires projecting *EAGLE* and Cylinder-Flow on a uniform rectangular grid. However, such a discretization scheme can not adapt its spatial resolution as a function of the geometry of the scene, which therefore constitutes a disadvantage with respect to an irregular triangular mesh. To limit this effect, the resolution of the grid is chosen such that the number of pixels is at least ten times larger than the number of points in the triangular mesh.

We project Cylinder-Flow onto a uniform $256 \times 64$ grid and *EAGLE* onto a $256 \times 128$ grid (the dimensions were chosen to respect the height-width ratio of the original data). The value of the pressure and velocity fields at each point in the grid is extrapolated from the nearest point in the raw simulated data. We illustrate this projection in figure 9. While the grid-based simulation (figure 9b) seems visually more accurate than the mesh-based simulation (figure 9d), we observed that the re-projection error (ie. the error obtained after projecting the grid based data onto the triangular mesh) is greater near sensible regions, as for example near the scene boundaries.

## B MODEL DETAILS

### B.1 CLUSTERING

We use our own implementation of the *same size Kmeans* algorithm described here[2]. Using equally sized clusters has two main advantages :

---

[2]https://elki-project.github.io/tutorial/same-size_k_means

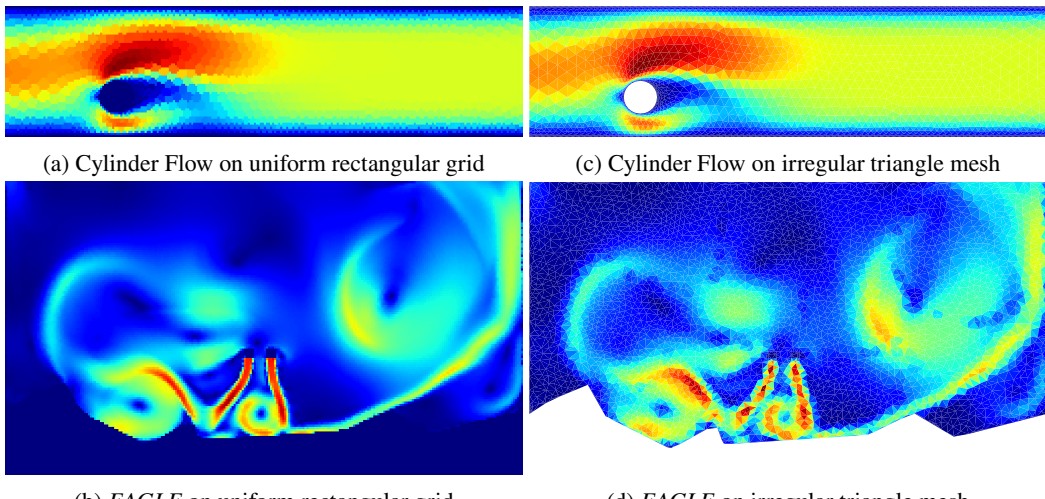

(a) Cylinder Flow on uniform rectangular grid     (c) Cylinder Flow on irregular triangle mesh

(b) *EAGLE* on uniform rectangular grid     (d) *EAGLE* on irregular triangle mesh

Figure 9: Illustration of the pixellisation process. The left column (a and b) shows snapshots of simulations from the grid-based datasets, used to train DilResNet. For comparison, we show the same snapshots in the mesh-based datasets (c and d). While resolution seems better on grid-based simulation, it lacks precision near sensible region, which are primordial for accurate forecasts.

- Areas of high density will be covered by a greater number of clusters, allowing the adaptive resolution of irregular meshes to be preserved on the coarser mesh.

- The model can be implemented efficiently, maximizing parallelization, since clusters can be easily stored as batched tensors.

Since the clustering depends solely on the geometric properties of the mesh (and not on the prediction of the neural network), it is possible to apply the clustering algorithm as a pre-processing step to reduce the computational burden during training. Note that since the mesh is dynamic, so are the clusters: the $k^e$ cluster at time $t$ will not necessarily contain the same points at time $t + 1$.

### B.2 ARCHITECTURE AND TRAINING DETAILS

We kept the same training setup for all datasets and trained our model for 10,000 steps with the Adam optimizer and a learning rate of $10^{-4}$ to minimize equation 6 with $\alpha = 10^{-1}$ and $H = 8$. Velocity and pressure are normalized with statistics computed on the train set, except for Scalar-Flow, where better results are obtained without normalization.

**Encoder** – $\phi_{\text{node}}$ and $\phi_{\text{edge}}$ are one-layer MLPs with ReLU activations, hidden size and output size of 128 (($\eta_i, e_{ij}) \in \mathbb{R}^{128}$). We used $L{=}4$ chained graph neural network layers composed of two identical MLP $\psi_{\text{edge}}$ and $\psi_{\text{node}}$ with two hidden layer of dimension 128, ReLU activated, followed by layer normalization. The positional encoding function $F$ is defined as follows:

$$F(x) = [\cos(2^i \pi x) \ \sin(2^i \pi x)]_{i=-3,\ldots3} \tag{8}$$

where $x$ is a 2D vector modeling the position of node $i$.

**Graph Pooling** – we used a single layer gated recurrent unit with hidden size of dimension $W$ followed by a single layer MLP with hidden and output size of $W$. This step produces a cluster feature representation $w_k \in \mathbb{R}^W$. For Cylinder-Flow and *EAGLE*, $W{=}512$. For Scalar-Flow, $W{=}128$.

**Attentional module** – Following (Xiong et al., 2020) an attention block is defined as follows for an input $w \in \mathbb{R}^W$ :

$$w_1 = \text{LN}(w) | F(\bar{x}_k)$$
$$w_2 = \text{MHA}(w_1, w_1, w_1)$$
$$w_3 = w + \text{Linear}(w_2)$$
$$w_4 = \text{LN}(w_3)$$
$$w_5 = \text{MLP}(w_4)$$
$$w_6 = w_3 + w_5$$

where LN are layer norm functions, Linear is linear function (with bias), MHA is multi-head attention and MLP is a multi-layer perceptron with one hidden layer of size $W$. We denote the barycenter of cluster $k$ as $\bar{x}_k$. We used $M=4$ chained attention block, with four attention head each. The last attention layer is followed by a final layer norm.

**Decoder** – The decoder takes as input the node embeddings $\eta_i$, the cluster features updated by the attentional module $w_k^M$ and the node-wise positional encoding $f_i$. We applied a graph neural network composed of two identical MLP (two hidden layers with hidden size of 128, ReLU activated and layer norm). The resulting node embeddings are fed to a final MLP with two hidden layers and hidden size of 128, with TanH as activation function.

### B.3 BASELINES TRAINING DETAILS

After performing a grid search to select the best options, we found that training each baseline to minimize equation 6 with Adam optimizer and learning rate of $10^{-4}$ produces best results. We vary the weighting factor $\alpha$ to maintain balance between pressure and velocity. For Cylinder-Flow and *EAGLE*, we trained the baselines over $H = 5$ time-steps. For Scalar-Flow, we set $H = 20$.

**MeshGraphNet** – we performed grid search over the number of GNN layers to fit to each dataset, but best results were obtained with the recommended depth $L = 15$ for each dataset. Conversely to what is suggested in Pfaff et al. (2021), we found that training MeshGraphNet over a longer horizon improves the general performances. We used our own implementation of the baseline and make sure to reproduce the results presented in the main paper (for Cylinder-Flow only). We get the best trade-off between velocity and pressure with $\alpha = 10$.

**GAT** – we performed a grid-search over the number of heads per layer and the number of layers. Best results were obtained for 10 layers of graph attention transformer and two attention heads per layer (except for Cylinder-Flow, where four heads slightly improves the performances).

**DilResNet** – we found that increasing the number of blocks improves overall performance, setting the number of convolutional blocks from 4 to 20.

The baselines are structurally built to predict pressure field $\hat{\mathcal{V}}'(t+h)$ and velocity field $\hat{\mathcal{P}}'(t+h)$ described on the mesh geometry at current time $\mathcal{N}(t)$. Auto-regressive forecasting on a longer horizon thus requires interpolation of the predicted flow to the (provided) future mesh $\mathcal{N}(t + h)$. We do not want interpolation to disturb our problem of interest, which is turbulent flow prediction. Therefore, we made the interpolation from $\mathcal{N}(t)$ to time $\mathcal{N}(t + 1)$ straightforward. As the vast majority of the mesh remains static (see previous section), only the nodes linked to the UAV need to be interpolated. Since they can readily be associated in a one-to-one relation, nearest point interpolation can be performed automatically by assigning $\hat{\mathcal{V}}'(t+h)$ at these points to $\hat{\mathcal{V}}(t+h)$.

## C   MORE RESULTS

### C.1 DETAILED METRICS

Formally, we used the following metrics to report our results on the test set $\mathcal{D}$:

$$\text{N-RMSE} = \frac{1}{H|\mathcal{D}|} \sum_{\mathcal{D}} \sum_{t=1}^{H} \frac{\|v(t) - \hat{v}(t)\|_2}{\tilde{v}} + \frac{\|p(t) - \hat{p}(t)\|_2}{\tilde{p}} \tag{9}$$

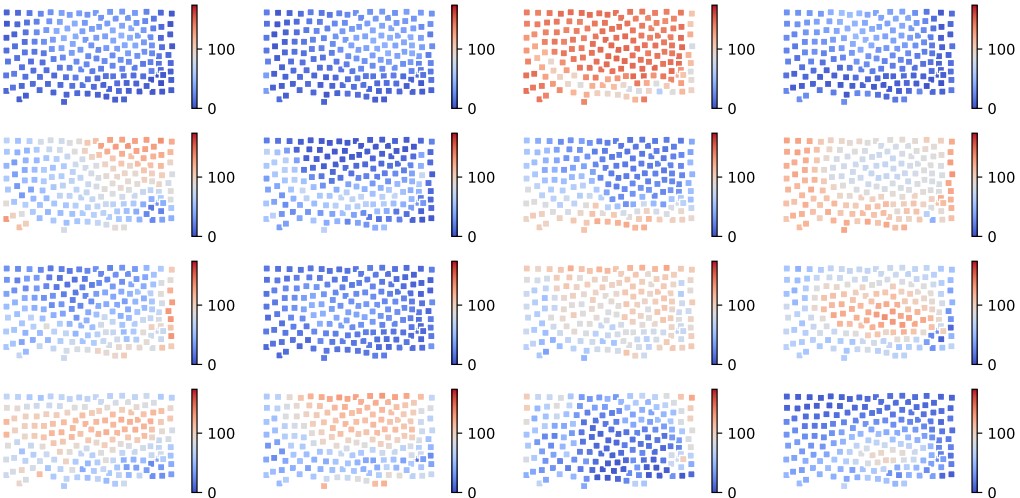

Figure 10: The maps show the k-number computed for each cluster, that is, the number of nodes required to reach 90% of the attention. A low k-number indicates a very specialized head (attending to few nodes), while a high k-number indicates uniform attention.

where $\tilde{v}$ and $\tilde{p}$ are standard deviation of velocity and pressure field computed on the train set.

**Detailed metric** – raw root mean squared error (RMSE) on each field is reported in figure 11 as well as temporal evolution of N-RMSE across prediction horizon. On Cylinder-Flow (Figure 11a), velocity error is very similar between MeshGraphNet and ours. Our model slightly outperforms the baseline on the pressure field, yielding overall better performances. However, temporal evolution of the N-RMSE indicated that both models converge to the same accuracy for very long roll-out prediction. On *EAGLE*, our model shows excellent stability over long horizon, and produces accurate velocity and pressure estimates.

**K-number** – is a property which can be calculated for attention maps, and which consists in the number of tokens required to reach 90% of attention (Kervadec et al., 2021). This property can be used to characterize the shape of attention maps, varying from peaky attention (requiring few tokens to reach 90%) to more uniform attention heads. We show k-numbers in Figure 10. Interestingly, the k-number maps can be compared with attention maps figure 5d: peaky heads (in blue) are correlated with relatively local attention maps, and conversely, more uniform heads (in red) correspond to attention maps focusing on larger distances, often following the airflow. Some heads have different behavior depending on the selected cluster, and are peaky in some areas (mainly around the boundaries of scene), but more uniform elsewhere. These cues support the importance of global attention in our model.

| Horizon | +1 | | +50 | | +250 | |
|---|---|---|---|---|---|---|
| Field | V | P | V | P | V | P |
| **MGN** | 0.0004 | 0.0016 | 0.0047 | 0.0095 | **0.0144** | 0.0145 |
| **GAT** | 0.0015 | 0.0025 | 0.0278 | 0.0360 | 0.2595 | 0.2314 |
| **DRN** | 0.0098 | 0.0063 | 0.0152 | 0.0085 | 0.0344 | 0.0152 |
| **Ours** | **0.0003** | **0.0007** | **0.0044** | **0.0035** | 0.0179 | **0.0079** |

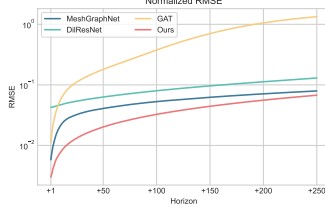

(a) **CylinderFlow**: (Right) RMSE on velocity **V** and pressure **P** fields. (Left) Normalized RMSE over forecasting horizon. Our mesh transformer overcomes the baselines by a small margin. Yet qualitative results tends to indicates that Cylinder Flow is already a well mastered task.

| Horizon | +1 | | +50 | | +100 | |
|---|---|---|---|---|---|---|
| Field | V | D | V | D | V | D |
| **MGN** | 0.0009 | 0.0059 | 0.0105 | 0.0568 | 0.0231 | 0.1130 |
| **GAT** | 0.0009 | 0.0066 | 0.0097 | 0.0578 | 0.0200 | 0.1091 |
| **DRN** | 0.0014 | 0.0101 | 0.0130 | 0.0750 | 0.0237 | 0.1217 |
| **Ours** | **0.0005** | **0.0024** | **0.0035** | **0.0145** | **0.0059** | **0.0239** |

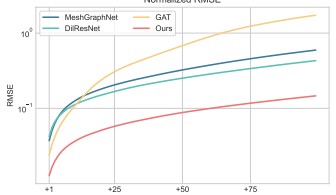

(b) **Scalar Flow**: (Right) RMSE on velocity **V** and density **D** fields. (Left) Normalized RMSE over forecasting horizon. Our model shows improvements over the baselines on both fields.

| Horizon | +1 | | +50 | | +250 | |
|---|---|---|---|---|---|---|
| Field | V | P | V | P | V | P |
| **MGN** | 0.0810 | **0.4256** | 0.5926 | 2.2492 | 1.0702 | 3.7220 |
| **GAT** | 0.1698 | 64.546 | 0.8551 | 162.56 | 1.0959 | 227.20 |
| **DRN** | 0.2517 | 1.4453 | 0.5374 | 2.4568 | 0.9188 | 3.5824 |
| **Ours** | **0.0537** | 0.4590 | **0.3494** | **1.4432** | **0.6826** | **2.4130** |

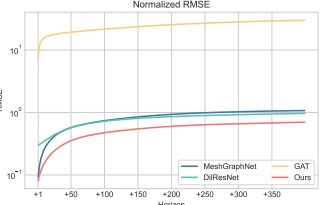

(c) *EAGLE*: (Right) RMSE on velocity **V** and pressure **P** fields. (Left) Normalized RMSE over forecasting horizon. Our largely and consistently and reliably outperforms the competing baselines. While MeshGraphNet and DilResNet shows comparable performances during first time-steps, our model succeed to control error accumulation for reasonable horizons and eventually presented better simulations.

Figure 11: **Detailed metrics** on Cylinder-Flow, Scalar-Flow and *EAGLE*, evaluated for each baselines and our model.

## C.2 QUALITATIVE RESULTS

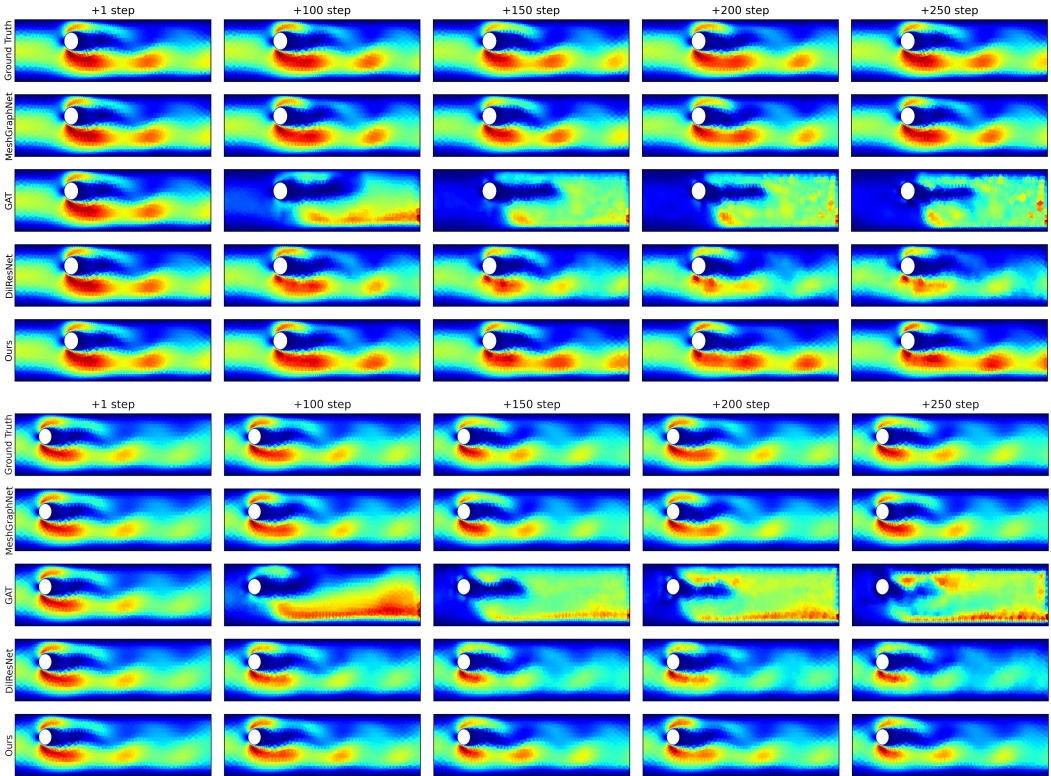

Figure 12: Examples of prediction forward in time on Cylinder-Flow

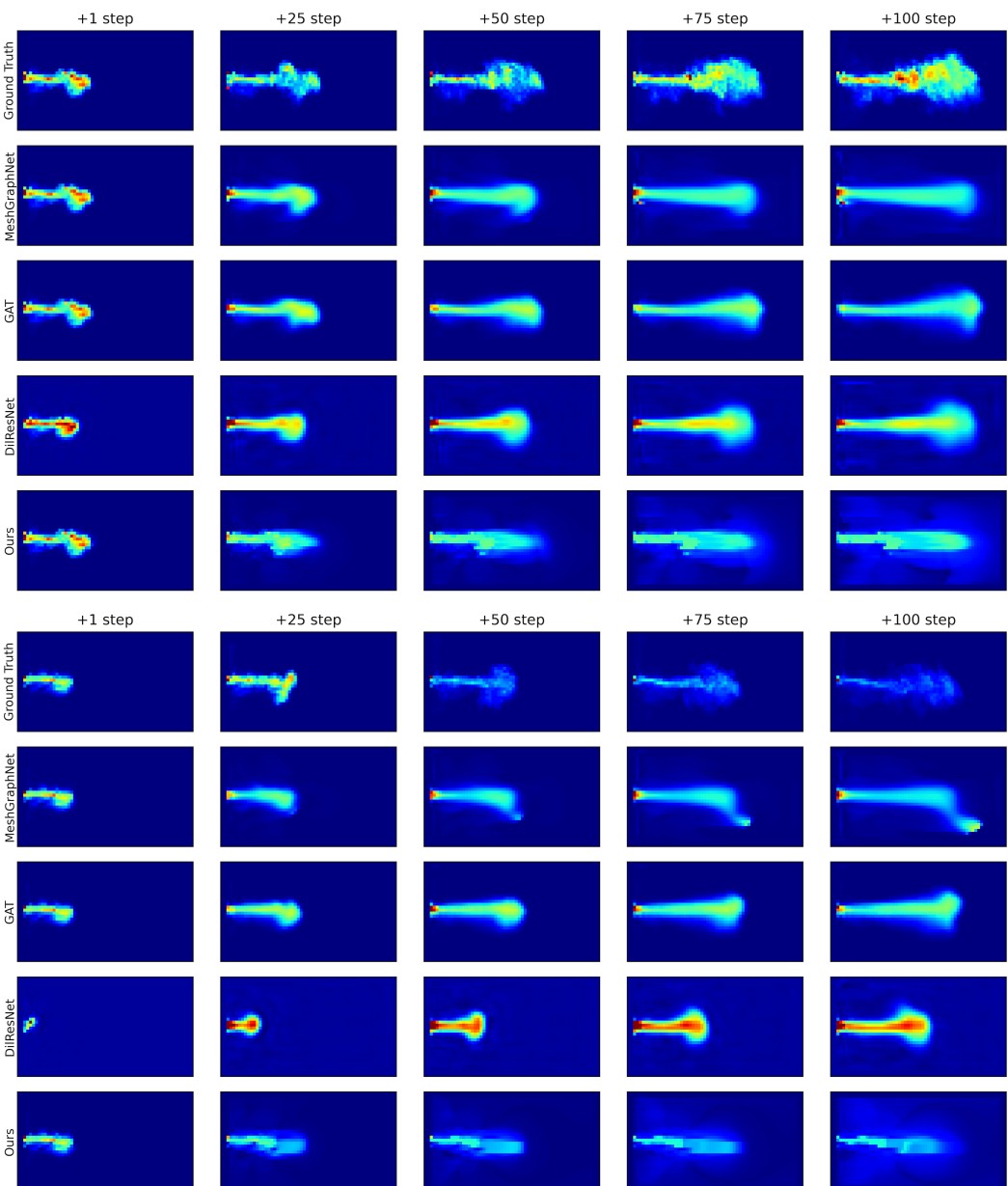

Figure 13: Examples of prediction forward in time on Scalar-Flow

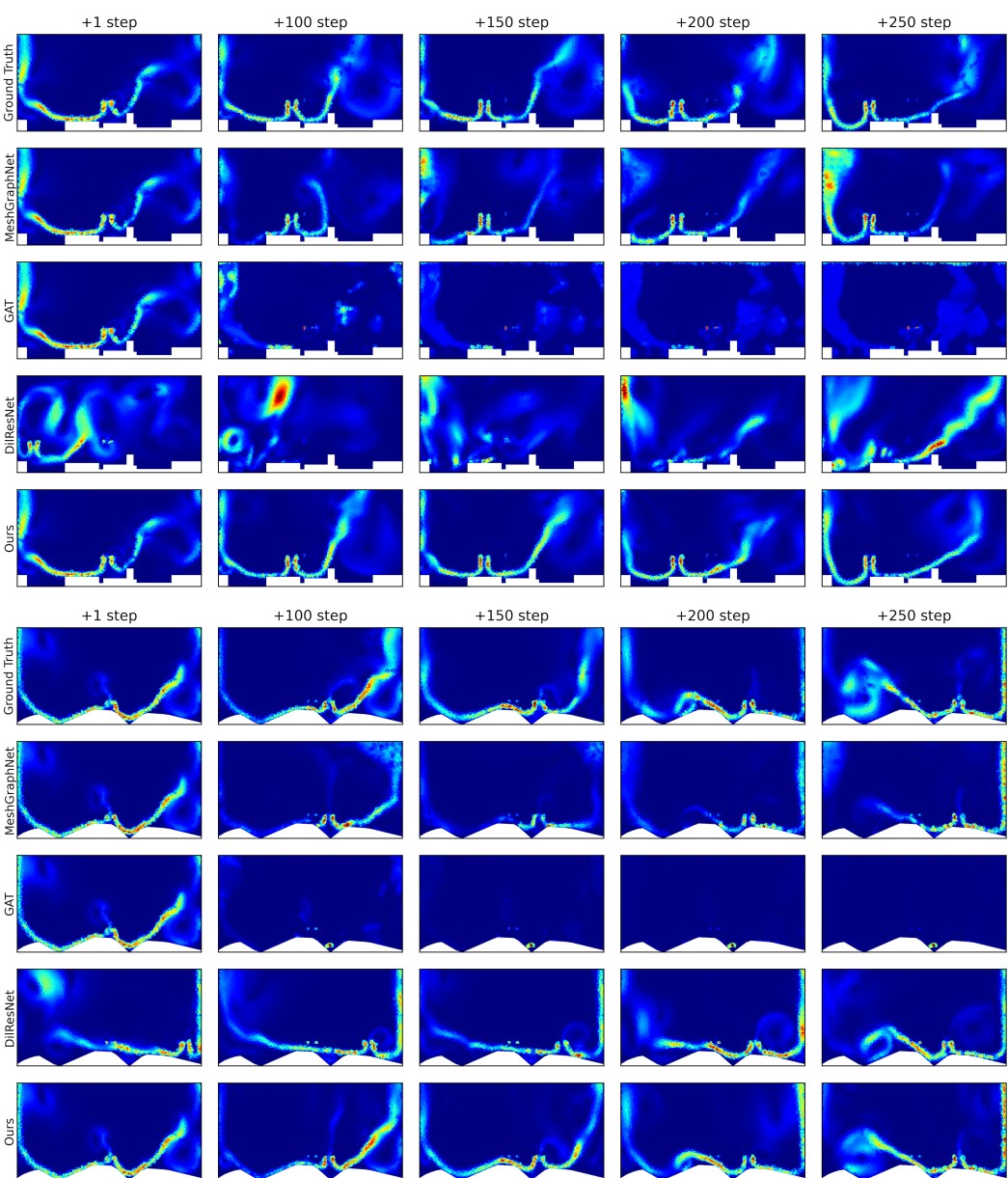

Figure 14: Examples of prediction forward in time on *EAGLE*

### C.3 FAILURE CASE

Despite the excellent performance of our model against competitive baselines, there is still room for improvement. Some more difficult configurations give rise to very turbulent flows, widely extended in the scene. The evolution of these flows is more difficult to predict and the models we evaluated failed to remains accurate. In these cases, the precision with which the small vortices are simulated is essential, because some of them will grow to become the majority.

Moreover, our model suffers from an error accumulation problem, like any auto-regressive model. Experimentally, we observe that the airflow tends to be smoothed by deep learning models when the prediction horizon increases.

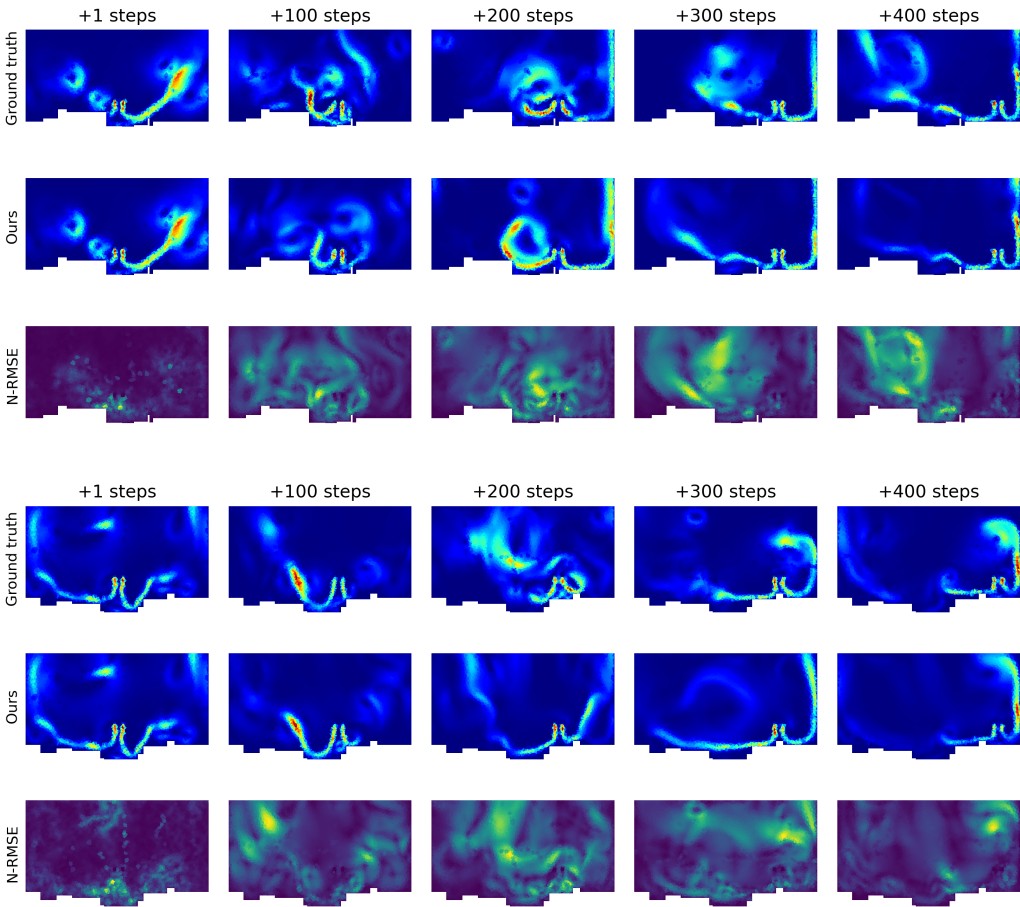

Figure 15: We expose failure cases of our mesh transformer on Eagle. The error increases when the flow tends to intensify throughout the scene, and when turbulence dominates. Over a longer prediction horizon, the airflow tends to be smoother and less turbulent.

| N-RMSE | | Training | | |
|---|---|---|---|---|
| | | 90% | 80% | 70% | 60% |
| Testing | 90% | **0.454** | 0.497 | 0.513 | 0.502 |
| | 80% | **0.427** | 0.446 | 0.467 | 0.440 |
| | 70% | 0.406 | 0.405 | 0.416 | **0.394** |
| | 60% | 0.401 | 0.370 | 0.368 | **0.348** |

Table 4: **Mesh down-sampling**: We train our mesh transformer under different regimes of down-sampling by keeping a fix percentage of points from the initial mesh and removing the others. We evaluate the resulting models on regimes different from training, and observe very little variations in N-RMSE among them.

### C.4 GENERALIZATION TO DIFFERENT MESH RESOLUTION

In *EAGLE*, the number of points varies from one simulation to another, forcing the model to generalize on meshes of different sizes. We explicitly demonstrate the performance of our mesh transformer on this task in table 4. Four instances of the model are trained on a particular regime in which the simulation meshes are randomly down-sampled, respectively at 90%, 80%, 70% and 60% of the initial mesh resolution. These models are then evaluated in a different regime from the one used for training, either higher (more points on average during test than during training), or lower (fewer points in test than in training). We show that our model generalizes well to these different regimes by giving relatively close N-RMSE measurements for a given down-sampling regime.

## D EXTENSION TO 3D FLUID SIMULATION

While we think that 3D simulations are indeed the long-term future on this subject, we argue that the complexity in factors of variation we need for large-scale machine learning is currently not possible in 3D simulations, and this has motivated our choice for a challenging 2D dataset. In this section, we discuss the possible extension of our dataset and the mesh transformer method to problems in three dimensions. We address two aspects: data generation itself, and the extension of the method.

### D.1 DATA GENERATION

Fluids datasets in 3D are very limited, due to the computation time required for simulation. Mesh-based simulation in three dimensions greatly increases the number of points in the mesh, and thus exponentially increases the computing time (see numerical evidences in Kim (2019); Dantan et al. (2017)). Classical workarounds rely on relaxing physical accuracy or versatility of the solver, e.g. with SPH simulations. Accurate 3D simulations are mostly conducted on grid-based meshes, and for rather simple, theoretic problems (Mohan et al., 2020a; Chen et al., 2021b; Stachenfeld et al., 2021). The John Hopkins Turbulent Database (Li et al., 2008) contains nine direct numerical simulation datasets (i.e. direct resolution of Navier-Stokes equations) but with only a single scene per dataset simulated on a very fine grid and low time resolution.

Therefore, extending *EAGLE* to 3D simulations is very difficult without sacrificing one of the fundamental principles on which our dataset relies: **(i) accuracy**, guaranteed by the resolution of RANS equations with demanding turbulence model on a very fine mesh **(ii) irregular meshes**, which are much more versatile and widespread in engineering and **(iii) large scale**, with nearly 1200 different scene configurations.

### D.2 MODELS AND METHODS

Forecasting models in the literature mostly focus on 2D simulations (Li et al., 2019; Kashefi et al., 2021; Han et al., 2022; Thuerey et al., 2020). To the best of our knowledge, there is no published work on large-scale machine learning models performing flow prediction on irregular 3D meshes. Therefore, publishing a 3D version of *EAGLE* seems premature, not to mention the difficulty of distributing such a dataset and training models on reasonable setups. For grid-based simulation on the other hand, few works leverage CNN-like structures for flow prediction (Stachenfeld et al., 2021;

Mohan et al., 2020b; Fonda et al., 2019) or computer graphics Wiewel et al. (2020); Chu & Thuerey (2017), yielding the limitations discussed in the main paper.

However, we emphasize that GNN-based model are theoretically not restricted to 2D and can readily manage 3D simulations. The main challenge is the memory requirements to train graph neural networks on larger point-clouds. We argue that our mesh transformer is a step towards handling larger meshes by reducing the number of features using clustering

