# OpenReview forum: "EAGLE: Large-scale Learning of Turbulent Fluid Dynamics with Mesh Transformers"
_ICLR.cc/2023/Conference — ICLR 2023 poster_

### Official Review · Reviewer_jvKK · 2022-10-23

**Confidence:** 3
**Correctness:** 3
**Technical Novelty And Significance:** 2
**Empirical Novelty And Significance:** 3
**Recommendation:** 5

**Clarity, Quality, Novelty And Reproducibility:**

There are quite a few things unclear to me in the simulation dataset:
- Are there any reasons behind the choice of simulating RANS instead of the original Navier-Stokes equations in the dataset?
- It looks like the drone and the fluid are one-way coupled instead of two-way coupled: the drone provides boundary conditions to the fluid’s motion, but the fluid does not affect the drone’s motion (there are no velocity or pressure terms in Eqn. 8). If the fluid has no influence on the drone, I feel the MPC control is a bit redundant and a kinematic drone should suffice.
- I am not sure I get how the drone’s mesh serves as the boundary condition of the fluid area. It looks like its bounding box is used. Could you elaborate on the velocity/pressure boundary conditions around the bounding boxes? I was expecting some inlet/outlet flow velocity to be set at where the propellers are, but I didn’t find details about it.
- I don’t quite understand the task definition at the end of Sec. 3. Do the node positions x_i change over time? It looks like the fluid area is remeshed at every frame if I am not mistaken.

I initially thought the proposed dataset simulated two-way coupled Navier-Stokes fluids and a dynamic, controllable UAV when reading the main text and was quite excited. It would be good if the paper could clarify the setting upfront in Sec. 3.

Regarding the network model, it looks like the velocity and pressure fields are predicted purely based on the original mesh at the current time followed by resampled to the new mesh if I am not mistaken. This seems questionable to me because I think the correct velocity and pressure field at the next time step will be determined by both the current and new mesh.

Reproducibility: I guess this largely depends on what will be included in the released dataset. Replicating the full simulation results using Ansys is probably intractable for people with limited access to computational resources, but I still suggest the dataset include configurations of the scenes in Ansys if possible.

Other comments:
- Is EAGLE an acronym?
- The baselines are confusing: the beginning of Sec. 5 refers MeshGraphNet and Cylinder-Flow to Li et al. 2019, but I don’t think that paper means either of them. By MeshGraphNet the paper probably means Pfaff et al. [2021].
- I appreciate that the paper frankly talked about failure cases and poor results (Scalar flow) in the experiments.


**Strength And Weaknesses:**

The proposal of a large-scale fluid simulation dataset is very timely. I believe having a high-quality dataset will foster relevant research in the learning community.

On the other hand, part of me feels this dataset is still premature. It is probably better than many existing fluid simulation datasets in previous papers, but I think it could have been more polished.


**Summary Of The Paper:**

This paper makes two contributions: First, it presents a large-scale 2D fluid simulation dataset called EAGLE. The dataset consists of 2D scenes with (dynamically changing) rigid boundaries and Reynold-Averaged Navier-Stokes (RANS) fluids. Second, it proposes a neural network model (mesh transformer) for solving fluid problems and compares its performance with popular neural network simulators (MeshGraphNet, etc).

**Summary Of The Review:**

Currently, I am leaning toward rejection, although my score is not final yet. It is definitely a respectable submission, and I welcome different opinions from the authors and other reviewers.

---

> ### Author Response · Authors · 2022-11-16
> **Response to Reviewer jvKK (1/2)**
>
> We thank the reviewer for their work.
>
> ### **Q -** Are there any reasons behind the choice of simulating RANS instead of the original Navier-Stokes equations in the dataset?
> **A -** Direct Numerical Simulation (DNS, i.e. solving directly Navier-Stokes equation) is not possible for our purpose (Veersteg & Malalasekera 2007). Solver convergence requires **extremely fine-grained meshes**, down to Kolmogorov scale (roughly 41 millions points for Eagle) which is absolutely intractable for our task.
>
> Note that direct solvers of Navier-Stokes equations are rare in practice: one exception is the John Hopkins Turbulence Dataset. However, it only handles very simple setups and is limited to a single simulation on a squared grid. A single simulation in this database represents 5 to 130 Terabytes. Assuming proportionality between computation time and number of points (the actual relation is super linear), **the entire simulation of Eagle in DNS would have required about 85 years** of simulations on 8x A100 GPUs in parallel.
>
> For this reason, RANS is a very natural tool in our context. It is commonly used for several equations in CFD, in engineering and research, and the Fluent software is highly used in aerodynamic design (cars, aircraft etc.).
>
>
> ### **Q -** It looks like the drone and the fluid are one-way coupled instead of two-way coupled: the drone provides boundary conditions to the fluid’s motion, but the fluid does not affect the drone’s motion (there are no velocity or pressure terms in Eqn. 8). If the fluid has no influence on the drone, I feel the MPC control is a bit redundant and a kinematic drone should suffice [...] I initially thought the proposed dataset simulated two-way coupled Navier-Stokes fluids and a dynamic, controllable UAV when reading the main text and was quite excited. It would be good if the paper could clarify the setting upfront in Sec. 3.
>
> **A -** The coupling is indeed mono-directional, we remove ambiguity in section 3 by mentioning explicitly that the dynamical model used for MPC is flow-agnostic. However, our intention was never to create a bi-directional coupling, which we argue would not provide much added value to the purpose of Eagle: our dataset is dedicated to **representing challenging dynamical behaviors in CFD**, e.g. turbulence (see Table 1, and beginning of section 3), and using mono- or bi-directional coupling does not make a big difference for this purpose. The purpose of the MPC controller was to create realistic drone trajectories and control inputs, but we agree that simpler handcrafted trajectories would have done a similar job.
>
> Moreover, the practical interest of the bi-directional coupling to our contribution is not clear, apart from the technical challenge of the implementation. Our paper does not focus on drone control. While work on studying the effect on UAV control exists (see references below), these experiments require learning from interactions directly using a simulator or a real drone, and cannot be done on a static dataset. It is unlikely that such a coupling will change the **complexity of the airflow in the simulation**.
> The dynamic drone control is a minor feature of the dataset. Our main focus is (a) the superior difficulty of our dataset compared to existing works, (b) the quality of the simulations and (c) the proposal of a new model that beats the state of the art on several tasks of fluid dynamics, including ours.
>
> References:
> - Shi, Guanya, et al. "Neural lander: Stable drone landing control using learned dynamics." Conference on Robotics and Automation (ICRA). IEEE, 2019.
> - Bauersfeld, L., Kaufmann, E., Foehn, P., Sun, S., & Scaramuzza, D. (2021). Neurobem: Hybrid aerodynamic quadrotor model. Robotics: Science and Systems (RSS).
>
> ### **Q -** I am not sure I get how the drone’s mesh serves as the boundary condition of the fluid area. It looks like its bounding box is used. Could you elaborate on the velocity/pressure boundary conditions around the bounding boxes? I was expecting some inlet/outlet flow velocity to be set at where the propellers are, but I didn’t find details about it.
>
> **A -** We updated the appendix to state this more clearly (see A.1, simulation details). Bounding boxes are simply a way to obtain **accurate mesh points near the drone**. This is done to ensure a sufficient number of points around the moving drone. As mentioned in the appendix:
>
> "_we used the ground truth trajectory to track drone position and orientation across time and extrapolate bounding boxes, which are then transformed into point clouds by subdividing the box into several points_"
>
> Points on the drone boundaries are **labeled** as wall nodes (for left, right, and top points), and inlet (for bottom points). We use canonical boundary conditions, that is: no-slip condition (zero velocity) for walls, zero diffusion of flow variables on outlet, and normal velocity at inlet (velocity is proportional to the propeller speed obtained via MPC).

---

> > ### Author Response · Authors · 2022-11-16
> > **Response to reviewer jvKK (2/2)**
> >
> > ### **Q -**  I don’t quite understand the task definition at the end of Sec. 3. Do the node positions x_i change over time? It looks like the fluid area is remeshed at every frame if I am not mistaken.
> >
> > **A -** The fluid domain is indeed **remeshed at each time step**. However, the prediction of the dynamics of the mesh $\mathcal{N}(t+h)$ is outside the scope of this paper which rather focuses on the evolution of the physical variables. The task as described at the end of section 3 gives access to the initial conditions (mesh and physical variables at the initial time), as well as to the future geometry of the mesh $x_i(t+h)$. Only velocity and pressure need to be predicted by the model. We tried to make this clearer in the task definition.
> >
> > ### **Q -**  Regarding the network model, it looks like the velocity and pressure fields are predicted purely based on the original mesh at the current time followed by resampled to the new mesh if I am not mistaken. This seems questionable to me because I think the correct velocity and pressure field at the next time step will be determined by both the current and new mesh.
> >
> > **A -** We recall that future mesh positions are assumed to be accessible in our task definition. However, this involves two different phenomena:
> > - **We do not want flow information to leak through the mesh**: we specifically took care to NOT embed any flow information in the mesh geometry, to prevent shortcut and bias in the model. We state this in the appendix:
> >
> > "_Better mesh simplification algorithm exists, notably minimizing the interpolation error, yet such algorithms rely on the simulated flow to compute the mesh, which may embed unwanted biases or shortcuts in the mesh geometry_"
> >
> > Thus, future mesh positions are not mandatory for forecasting pressure and velocity fields.
> >
> >  - **We do need to interpolate the flow on future mesh**: due to the actual structure of the mesh (cf. appendix A.1, and website), the interpolation is straightforward : most points are static, except for the nodes linked to the UAV. Yet, since time resolution is small, these nodes actually move very little from a timestep to another. Then, flow interpolation to the nearest node is automatic and implicit. We updated appendix B.3 to state this explicitly.
> >
> > However, we mention that our choice of simplifying the interpolation to the new mesh has been driven by existing SOTA approaches : GNN-based **baselines are not designed for dynamical meshes**, and suffer from interpolation in a completely dynamical mesh. Conversely our mesh transformer could readily take into account new mesh by leveraging positional encoding from the future mesh in the Decoder.
> >
> > ### **Q -**  Reproducibility: I guess this largely depends on what will be included in the released dataset. Replicating the full simulation results using Ansys is probably intractable for people with limited access to computational resources, but I still suggest the dataset include configurations of the scenes in Ansys if possible.
> >
> > **A -** Ansys project files represent an additional 10Go of data that can be added to the simulations. **We will include these files** for better reproducibility. As mentioned in the paper, the complete dataset as well as the code of the method will be made publicly available.
> >
> > ### **Q -**  Is EAGLE an acronym?
> > **A -** When we selected the name EAGLE, an easy mapping to the title as an acronym was not optimized, but rather a relatable name, easy to remember, and conveying the link to a drone flying over land. We did, however, at this time look for a coherent (but, admittedly, convoluted) acronym: Eagle relates to “datasEt of lArGe scaLe dronE simulations”.
> >
> > ### **Q -** The baselines are confusing: the beginning of Sec. 5 refers MeshGraphNet and Cylinder-Flow to Li et al. 2019, but I don’t think that paper means either of them. By MeshGraphNet the paper probably means Pfaff et al. [2021].
> > **A -** Thank you very much for your proofreading, we are sorry for this mistake. We fixed it in the paper.

---

> > > ### Comment · Reviewer_jvKK · 2022-11-26
> > > **Thank you for the update**
> > >
> > > Thank you very much for the detailed response! After carefully reviewing your response and discussions with reviewers in other threads, I am still leaning toward rejection and plan to keep my score unchanged.
> > >
> > > This is definitely a respectable submission, and I am really looking forward to seeing a good paper from it in the future. However, when evaluating such a dataset, I personally favor physical plausibility over scale, i.e., I would be more excited if this dataset simulated a 3D, two-way coupled Navier-Stokes fluid and a dynamic, controllable UAV even if its dataset size had to be an order of magnitude smaller than what it currently presented.
> > >
> > > Alternatively, the submission does not have to be tied to drones at all. I agree with the authors that the presented dataset is more complicated than the cylinder examples that many other machine-learning papers typically use these days, so maybe it would be helpful to completely get rid of the drone part and reorganize the whole story to something more precisely reflecting what the simulation model achieves.

---

### Official Review · Reviewer_gcRV · 2022-10-24

**Confidence:** 3
**Clarity, Quality, Novelty And Reproducibility:** See above.
**Correctness:** 3
**Technical Novelty And Significance:** 3
**Empirical Novelty And Significance:** 3
**Recommendation:** 6

**Strength And Weaknesses:**

Strength:
1. Although different shapes of the boundary affect how the mesh is constructed and how they are clustered, it seems that the model still generalizes well.
2. The results are promising and supportive of the technical contribution.

Weakness:
1. Missing reference: Takahashi, Tetsuya, et al. "Differentiable fluids with solid coupling for learning and control." Proceedings of the AAAI Conference on Artificial Intelligence. Vol. 35. No. 7. 2021.
2. I wonder how this can be applied to 3D cases. Since the 2D task only there is already 270Gb data, it is probably not practical to prepare a good amount of data for the model to converge and generalize. Also, dealing with tetrahedra instead of triangles may introduce unknown challenges. Please add more insights regarding the extension to 3D in the conclusion section.
3. How does the mesh resolution affect the training/test accuracy? If the training and test images are of different resolutions, will the model perform better or worse? It would be great to see such comparisons in the experiments.


**Summary Of The Paper:**

This paper introduces a dataset for predicting fluid turbulences and a method using a Transformer to learn the dynamics. It adds clustering and graph pooling before the attention block so that the receptive field is not constrained in a local neighborhood. Results show that the new method outperforms the previous works.

**Summary Of The Review:**

I would recommend weak reject because of the weaknesses described above. I am happy to change my score if the issues are addressed in the rebuttal.

---

> ### Author Response · Authors · 2022-11-16
> **Response to Reviewer gcRV**
>
> We would like to thank the reviewer for their work.
>
> ### **Q -** Missing reference: Takahashi, Tetsuya, et al. "Differentiable fluids with solid coupling for learning and control." Proceedings of the AAAI Conference on Artificial Intelligence. Vol. 35. No. 7. 2021.
> **A -** Thank you for pointing us to this work. We **added this reference to our related work section**, however, it addresses a fundamentally different objective. They release a grid-based differentiable simulator to train neural networks to control fluid through solid objects. In our work, we propose a mesh-based dataset for learning fluid dynamics. Moreover, the simulator uses Fast Fluid Dynamics (FFD) equations, which are easier to solve (and differentiate) but less precise than the Computational Fluid Dynamics equations used in Ansys Fluent to simulate Eagle.
>
> ### **Q -** I wonder how this can be applied to 3D cases. Since the 2D task only there is already 270Gb data, it is probably not practical to prepare a good amount of data for the model to converge and generalize. Also, dealing with tetrahedra instead of triangles may introduce unknown challenges. Please add more insights regarding the extension to 3D in the conclusion section.
> **A -** To address this remark, we added a discussion on extension to 3D of our dataset and our model. There are indeed many unresolved issues making the use of 3D simulation in machine learning intractable yet. The **dataset** storage is one of them, associated with the **tremendous computational cost**. Existing 3D datasets are grid-based and contain only a handful or even only a single simulation in a theoretical setup. They do not allow large-scale deep learning targeting generalization. This is out of the scope of this paper, and we pointed out the weaknesses of grid based simulations in the main paper.
>
> To the best of our knowledge, **there is no mesh-based large-scale 3D dataset publicly available**, and therefore there is no 3D turbulence prediction model either. Machine learning models also face hardware limitations. The exploding number of nodes in the mesh is difficult to handle even on modern GPUs.
>
>
> ### **Q -** How does the mesh resolution affect the training/test accuracy? If the training and test images are of different resolutions, will the model perform better or worse? It would be great to see such comparisons in the experiments.
>
> **A -** The nature of Eagle's simulations already requires **varying the resolution from one simulation to another** (between 2354 and 4348 nodes per simulation). This is also the case for CylinderFlow. To complete this study, we trained our mesh-transformer on undersampled versions of Eagle, by randomly removing some of the mesh points. We performed the same operation during the test phase and obtained the following results:
> |  |  |  | Training |  |  |
> |:---:|:---:|:---:|:---:|:---:|:---:|
> |  |  | 90% | 80% | 70% | 60% |
> |  | 90% | 0.454 | 0.497 | 0.513 | 0.502 |
> | Testing | 80% | 0.427 | 0.446 | 0.4671 | 0.440 |
> |  | 70% | 0.406 | 0.405 | 0.416 | 0.394 |
> |  | 60% | 0.401 | 0.370 | 0.368 | 0.348 |
>
> Our model was already able to **generalize on meshes of different sizes**, it is also robust to a more significant change in the point distribution between train and test.

---

> > ### Comment · Reviewer_gcRV · 2022-11-22
> > **Thanks**
> >
> > Thank you for the responses. I would like to increase my score to weak accept.

---

### Official Review · Reviewer_DQoT · 2022-10-25

**Confidence:** 3
**Clarity, Quality, Novelty And Reproducibility:** The manuscript is well-written. I fol…
**Correctness:** 3
**Technical Novelty And Significance:** 2
**Empirical Novelty And Significance:** 3
**Recommendation:** 6

**Strength And Weaknesses:**

Strength:
- The dataset is large-scale. It covers a big fraction of the interest of the physical datasets. Previous datasets are mostly steady-state simulations.
- The manuscript compared their method with several well-known baselines.

Weaknesses:
- In my opinion, the weaknesses are obvious. First, the dataset only contains 2D data, whose usefulness is suspicious due to the diffusion of fluid. I wonder if the authors can provide more insights on how they are useful in real engineering.
- Several previous datasets are working in CFD, e.g., http://turbulence.pha.jhu.edu/. How is the proposed dataset compared to theirs?
- The only variation in the dataset is the geometry. However, in reality, there are more descriptions of fluid (it is true even for air itself), e.g., the Reynolds number, the temperature, the boundary conditions, and the wind speed. I wonder how generalizable a network trained on this dataset can be.

**Summary Of The Paper:**

This manuscript proposed a method to train neural networks to predict turbulent fluid dynamics. For the neural architecture, this manuscript introduced the concept of a transformer into the fluid dynamics field, which outperforms previous methods due to long-range receptive fields. To support the training of the mesh transformer, this manuscript generates a large-scale 2D aerodynamics dataset using a controlled drone. The experiments showed the superior performance of their method over several different baselines.

**Summary Of The Review:**

The dataset is valuable to the community. However, I would like to see the authors justify my questions about the dataset. I am open to changing the score based on the response.

---

> ### Author Response · Authors · 2022-11-16
> **Response to Reviewer DQoT**
>
> We thank reviewer QDoT for their work.
>
> ### **Q -** In my opinion, the weaknesses are obvious. First, the dataset only contains 2D data, whose usefulness is suspicious due to the diffusion of fluid. I wonder if the authors can provide more insights on how they are useful in real engineering.
>
> **A -** As mentioned in the global answer, while we agree that 3D simulations are indeed the long-term future on this subject, we argue that the complexity in factors of variation we need for **large-scale machine learning** is currently not possible in **3D simulations**. This is corroborated in the properties of the 3D datasets currently available, which focus on precision, but which are hardly large-scale: they consist of a handful, sometimes a single simulation, as e.g. the JHTDB dataset you mentioned (more on this further below).
>
> 3D simulation requires significantly more points than 2D, which increases computation time. For this reason, common practice in engineering **relies as much as possible on 2D simulations** for the development of a design, which will ultimately be validated on a single, more demanding 3D simulation. However, this does not mean that 2D simulation is without interest: it remains extremely time-consuming, and there is a real demand from engineering for fast and accurate solver on 2D problems, which will speed up the development process.
>
> Large-scale 2D datasets like Eagle will also allow the field in machine learning based CFD to develop data-driven methods, which **generalize over different realizations** of different underlying factors of variation (in our case the scene geometry), an advantage that ML brings to the traditional methods used in physics.
>
> As a summary, existing work is limited to one of the following:
> 1) the model tackles 3D simulations, but on **grid-based datasets** (not applicable to a vast majority of problems) and containing only a handful of different simulations on theoretical setups
> 2) the model focuses on mesh-based problems, but limited to **simple datasets**, like cylinder flow or airfoil.
>
> We added a discussion on extension to 3D in appendix D. **Simulations in Eagle are highly turbulent and non-steady**, but also defined on an irregular mesh, which make our dataset close to expectations in engineering. Moreover, we show in fig.7 of the paper that our model reaches SOTA performances while maintaining low prediction time. We believe that our contribution can help speed up simulations in engineering, providing an immediate solution to 2D mesh-based problems.
>
> Could we ask to further elaborate on the statement “usefulness is suspicious due to the diffusion of fluid”? While convection dominates in the mean flow, diffusion acts in the turbulent vortices and is therefore not negligible.
>
> ### **Q -** Several previous datasets are working in CFD, e.g., http://turbulence.pha.jhu.edu. How is the proposed dataset compared to theirs?
>
> **A -** We voluntarily did not compare ourselves to JHTB, which is cited in the original paper in the caption of table 1:
>
> "_Smaller-scale datasets such as Li et al. (2008); Wu et al. (2017) have been excluded, as they favor simulation accuracy over size._"
>
> These datasets are at the **antipodes** of Eagle. JHTB contains a set of unique, highly accurate, grid-based simulations in a highly theoretical situation. Large-scale deep learning is impossible with this dataset.
>
> Eagle on the other hand  is a large-scale dataset containing more than a thousand simulations, **suiting engineering quality expectations**, inspired by a practical setup and defined on an irregular mesh. Our dataset is intended for the ML community aiming to accelerate CFD using data-driven methods. JHTB datasets are regularly cited in physics, but had a very low impact or usage in machine learning, as it is difficult to generalize from a few realizations of important factors of variation. We provide more details in appendix D.
> ### **Q -** The only variation in the dataset is the geometry. However, in reality, there are more descriptions of fluid (it is true even for air itself),  the Reynolds number, the temperature, the boundary conditions, and the wind speed. I wonder how generalizable a network trained on this dataset can be.
> **A -** While we agree that even more factors of variation could be present, this holds even more for existing models and datasets in the literature. To the best of our knowledge, Eagle is already the most **challenging mesh-based dataset publicly available**. The variability from one simulation to another greatly exceeds previous work, since our simulations do not evolve in steady-state or periodic flow, conversely to widely used cylinder flow and airfoil datasets. It is unclear how adding more sources of variation will be beneficial to the community.
> Beyond these considerations, assuming other parameters to be fixed seems reasonable given our proxy task: flying in a room creates little variation in temperature, Reynolds number, and wind speed.

---

> > ### Comment · Reviewer_DQoT · 2022-12-06
> > **Thank you for the answers**
> >
> > I appreciate the effort the authors paid for responses. I read carefully, but still not convinced about the 2D-3D gap problem. I will raise the score to 6 to reflect my evaluations based on the answers to my other 2 questions.

---

### Official Review · Reviewer_912F · 2022-10-27

**Confidence:** 3
**Clarity, Quality, Novelty And Reproducibility:** I believe that the overall quality of…
**Correctness:** 4
**Technical Novelty And Significance:** 3
**Empirical Novelty And Significance:** 3
**Recommendation:** 6

**Strength And Weaknesses:**

Strength:
+ The paper is well written. Algorithms are well explained and easy to understand.
+ The results are good, from both qualitative and quantitative perspective.
+ The experiments are comprehensive and well demonstrate the robustness of the method

Weakness:
- The dataset is simulated by some existing dynamic fluid simulator, which means that the predicted results are highly impact the the quality of the simulator.
- Since there's no real experiments has been carried out and the analysis are based on 2D, I am not sure how and how well this model can work in the 3D real scene scenarios. It will be great if the authors can briefly discuss its applications or include some real results.
- It will be great if there's a failure case analysis or discussion.

**Summary Of The Paper:**

This paper proposed a novel GNN architecture and benchmark dataset to analyze the 2D turbulent fluid dynamics caused by a single moving flow source. By utilizing the autoregressive model and self-attention modules, the proposed method achieved outperforming performance on the testing dataset.

**Summary Of The Review:**

In summary, I think this paper is well written and provide some solid results with respect to the 2D dynamic fluid analysis. I think it will be better if the authors can include a short paragraph discussing the real scene application and failure case discussion.

---

> ### Author Response · Authors · 2022-11-16
> **Response to Reviewer 912F**
>
> We thank reviewer 912F for their work.
>
> ### **Q -** The dataset is simulated by some existing dynamic fluid simulator, which means that the predicted results are highly impact the the quality of the simulator.
> **A -** **Experimental setups** for measuring flow variables in real world scenario are **laborious and expensive**, do not provide enough information to reconstruct the flow of all state variables in the domain without using alignment with a simulation (e.g. Scalar Flow). Most importantly, real world measurements are intractable for large scale datasets.
>
> The impact of the simulator on the results is a well known concern in CFD and taken into account in Eagle. Our goal is not to achieve exceptional precision, as it is of interest solely for a very advanced theoretical study of the turbulence created by a drone. We claim to provide a **dataset of numerical simulations reaching the high engineering standards**, which allow the community to step up from steady-state air flow to a much more challenging task. .
>
> We also notice that **the problem also exists for the other datasets** cited in the paper, which are all based on simulation. These datasets use either a less efficient uniform mesh (less suited for most applications), or a less precise turbulence model on easier setups. Eagle has been carefully simulated with Ansys Fluent, an extensively tested professional CFD software. The dynamics is simulated by RANS equations and a much more accurate turbulence model than those usually used (Reynolds-stress instead of k-$\epsilon$ / k-$\omega$), which leads to significant trust in the solving method.
>
> ### **Q -** Since there's no real experiments has been carried out and the analysis are based on 2D, I am not sure how and how well this model can work in the 3D real scene scenarios. It will be great if the authors can briefly discuss its applications or include some real results.
>
> **A -** It is extremely difficult to provide results on **real-world problems** since gathering enough measurements to train a ML model is expensive and difficult. Likewise, *there are no mesh-based 3D datasets in the literature*, mostly because of the massive increase in simulation time. Several works focused on 3D uniform grid datasets, but are outside the scope of our contribution: this is already solved by CNN networks and we have shown the limits of uniform grids in the paper.
>
> **Results on real or 3D datasets seem premature** given previous work: existing models capable of working on meshes are limited to very simple, simulated, 2d problems like CylinderFlow or Airfoil. Eagle is already a big leap in complexity for the community.
>
> However, we still believe that our contribution has a **direct application in engineering**, because the vast majority of simulations are actually conducted in 2D, again due to computation time. Our dataset and our model are relevant for this. We discuss the extension of our contributions to 3D in an added appendix D.
>
> ### **Q -** It will be great if there's a failure case analysis or discussion.
>
> **A -** As pointed out by reviewer jvKK, the main paper discussed the failure of every model on the challenging real world dataset ScalarFlow. We complete our analysis with a **failure case** illustrated in the added appendix C.3. We identified challenging instances in Eagle giving rise to very turbulent flows, widely extended in the scene. The evolution of these flows is more difficult to predict and the models we evaluated failed. In these cases, the precision with which the small vortices are simulated is essential, because some of them will grow to become the majority.
>
> Moreover, our model suffers from an **error accumulation problem**, like any auto-regressive model. Experimentally, we observe that the airflow tends to be smoothed by deep learning models when the prediction horizon increases.

---

> > ### Comment · Reviewer_912F · 2022-11-28
> > **Thanks**
> >
> > Thank you for the updates. After carefully going through all the rebuttal comments, I'd like to keep my original rating. Even though I understand that testing the algorithm on real or 3D scenarios will be extremely hard, I am not fully convinced that algorithms trained merely on synthesized 2D data can be well applied to 3D and solve real world applications.

---

### Author Response · Authors · 2022-11-16
**Common answer**

We would like to thank the reviewers for their efforts and their reviews. We are happy that they appreciated our work on a new mesh transformer architecture reaching SOTA against strong baselines in both our and existing datasets. One reviewer also highlights the extensive ablation study conducted on our model to provide insights of its behavior. We believe that the way our model dynamically adapts the attention maps to the flow is a very promising and important part of our contribution as it partially explains our success.

However, we think that the significant increase in complexity we added to the data & benchmarks available to the field, one of the contributions of this paper, has been overshadowed by the impact of the perception of a 2D drone on the reviewing process. We think (and this has been implicitly stated by one of the reviewers), that the ratings reflect what the reviewers optimally would have liked to have (and which is arguably not yet possible to deliver), instead of the contribution and the potential impact of this more advanced dataset on the field. Compared to existing datasets in the literature (_Pfaff et al. 2022_, _De Bezenac et al. 2018_, _Stachenfeld et al. 2021_ are popular comparable 2D datasets), Eagle is unique and advancing the field for several reasons:
- It is the **most challenging dataset of mesh-based simulations** publicly available, far beyond classic flow past cylinder and airfoil dataset
- It is **well aligned with current needs of the community**, as existing models are not ready to handle more complexity.
- Datasets of comparable or better accuracy and complexity are **grid-based**, tackle theoretical problems only and are of limited size. Eagle not only balances task complexity, practical usability, and scale, it pushes further the research on ML for fluid dynamics.

While we agree that 3D simulations are indeed the long-term future on this subject, we argue that the complexity in factors of variation we need for large-scale machine learning is currently not possible in 3D simulations. This is corroborated in the properties of the 3D datasets currently available, which focus on precision, but which are hardly large-scale: they consist of a handful, sometimes a single simulation, as e.g. the JHTDB dataset mentioned by reviewer DQoT.

The reviews have shed little light on our second contribution: we propose a new mesh-transformer achieving SOTA on both existing datasets and our dataset against competing baselines from the literature. We have tried to thoroughly demonstrate the merits of our approach through numerous ablations, highlighting
- the interest of **spatial attention**,
- the need for **clustering** to allow its application on large meshes and
- the **sensitivity to the size of the cluster**.
We also provide insights on why our model outperforms classic GNN baselines by leveraging a **wider receptive field**. We hope that reviewers will acknowledge the relevance of this part of our contribution.

We summarize below the changes in the paper:

- We added a “purpose” paragraph at the beginning of section 3
- Simulation details have been clarified in appendix A
- Interpolation as been made explicit at the end of appendix B
- We added a discussion about extending our work to 3D in appendix D.-
- Minor changes suggested by reviewers.
- We illustrate worst case scenarios in appendix C.3

We have answered specific concerns by reviewers individually, further below.

---

### Public Comment · ~Shuhao_Cao1 · 2023-02-02
**Some questions**

Nice paper, and it would be nice if the authors can clarify some of my confusion about the EAGLE dataset.

> Numerical simulations were carried out using the software Ansys© Fluent, which solves the Reynolds Averaged Navier-Stokes equations of the Reynolds stress model. It uses five equations to model turbulence, more accurate than standard $k$-$\epsilon$ or $k$-$\omega$ models (two equations).

Can the five equations be specified explicitly in the camera-ready version? It says Reynolds stress model which I assume that the velocity-pressure-stress formulation is used, but right before this paragraph, it says "velocity field as well as the pressure field" are simulated, this means actually the velocity (+fluctuation)-pressure formulation is used? May I know what discretization method Ansys Fluent is using? FEM or FVM (cell-centered or vertex-based)?

> The encoder also computes an appropriate positional encoding based upon spectral projection $F(x)$.

Is this $F(\cdot)$ the same $F$ defined in equation (8)?

---

> ### Author Response · Authors · 2023-02-07
> **Thank you**
>
> Thank you for your comment and we are glad that it has been of interest to you. We apologize for any confusion in our paper. Simulations are conducted with Ansys Fluent with the Reynolds stress viscous model using the Stress-omega setup with shear flow corrections, we will make sure to include these details in the camera-ready version. For theoretical details about how the simulation are handled in the solver, please reach for the ANSYS Fluent Theory Guide, especially section 9.4 for a thorough description of Reynolds Stress model as implemented by Ansys Fluent.
>
> The positional encoding $F(\cdot)$ used in our mesh transformer is indeed the function described in eq. 8.
>
> Thank you again for your questions and feedback.

---

### Decision · Program_Chairs · 2023-01-20

**Decision:**

Accept: poster

**Justification For Why Not Higher Score:**

See the Summary Of AC-reviewer Meeting.

**Justification For Why Not Lower Score:**

See the Summary Of AC-reviewer Meeting.

**Metareview: Summary, Strengths And Weaknesses:**

This paper proposes a new dataset (named EAGLE) for large-scale 2D dynamic fluid simulation and a MeSH Transformer model for solving fluid problems.  The proposed model incorporates clustering and graph pooling operations into Transformer, and obtained good supportive results (outperforming well-known baselines) on both existing datasets and the new EAGLE dataset.

The paper is well-written and easy to follow, the new dataset with large-scale 2D dynamic simulation covers a big fraction of the interest of the physical datasets, and the experimental results are comprehensive for supporting the technical contribution.

On the other hand, all the reviewers pointed out that the lack of 3D simulation in this paper makes it less exciting. Also, an enlarged coverage by 2D simulations is desirable, including the settings with varying Reynolds number, temperatures, boundary conditions, and more (Reviewers DQoT, jvKK). We encourage the authors to take the reviewers feedback into account for improving the current work.


**Note From Pc:**

if the above contains the word "oral" or "spotlight" please see: "oral" presentation means -> notable-top-5% and "spotlight" means -> notable-top-25%. As stated in our emails, we are disassociating presentation type from AC recommendations

**Summary Of Ac-Reviewer Meeting:**

The zoom meeting and email conversations among the AC and reviewers mainly focused on 1) whether 3D simulations would be a requirement for accepting this paper, and 2) whether the new dataset (2D) and the experiments in this paper add significantly new knowledge to the field.  The answer for the first item is that 3D simulation should not be a requirement but the lack of 3D makes the current paper less exciting.  As for the 2nd point, we feel that the new 2D dataset is useful and their experiments are informative but more settings could or should be explored.

In conclusion, this is truly a borderline case. We recommend acceptance with low confidence. That is, if there are other obviously stronger  papers across different areas, a rejection of this paper is also an OK option.